# Improved Regret for Bandit Convex Optimization with Delayed Feedback

**Yuanyu Wan**[1,2,3], **Chang Yao**[1,2,3], **Mingli Song**[2,3], **Lijun Zhang**[4,2]

[1]School of Software Technology, Zhejiang University, Ningbo, China
[2]State Key Laboratory of Blockchain and Data Security, Zhejiang University, Hangzhou, China
[3]Hangzhou High-Tech Zone (Binjiang) Institute of Blockchain and Data Security, Hangzhou, China
[4]National Key Laboratory for Novel Software Technology, Nanjing University, Nanjing, China
{wanyy,changy,brooksong}@zju.edu.cn, zhanglj@lamda.nju.edu.cn

## Abstract

We investigate bandit convex optimization (BCO) with delayed feedback, where only the loss value of the action is revealed under an arbitrary delay. Let $n, T, \bar{d}$ denote the dimensionality, time horizon, and average delay, respectively. Previous studies have achieved an $O(\sqrt{n}T^{3/4} + (n\bar{d})^{1/3}T^{2/3})$ regret bound for this problem, whose delay-independent part matches the regret of the classical non-delayed bandit gradient descent algorithm. However, there is a large gap between its delay-dependent part, i.e., $O((n\bar{d})^{1/3}T^{2/3})$, and an existing $\Omega(\sqrt{dT})$ lower bound. In this paper, we illustrate that this gap can be filled in the worst case, where $\bar{d}$ is very close to the maximum delay $d$. Specifically, we first develop a novel algorithm, and prove that it enjoys a regret bound of $O(\sqrt{n}T^{3/4} + \sqrt{dT})$ in general. Compared with the previous result, our regret bound is better for $d = O((n\bar{d})^{2/3}T^{1/3})$, and the delay-dependent part is tight in the worst case. The primary idea is to decouple the joint effect of the delays and the bandit feedback on the regret by carefully incorporating the delayed bandit feedback with a blocking update mechanism. Furthermore, we show that the proposed algorithm can improve the regret bound to $O((nT)^{2/3}\log^{1/3}T + d\log T)$ for strongly convex functions. Finally, if the action sets are unconstrained, we demonstrate that it can be simply extended to achieve an $O(n\sqrt{T\log T} + d\log T)$ regret bound for strongly convex and smooth functions.

## 1 Introduction

Online convex optimization (OCO) with delayed feedback [Joulani et al., 2013, Quanrud and Khashabi, 2015] has become a popular paradigm for modeling streaming applications without immediate reactions to actions, such as online advertisement [McMahan et al., 2013] and online routing [Awerbuch and Kleinberg, 2008]. Formally, it is defined as a repeated game between a player and an adversary. At each round $t$, the player first selects an action $\mathbf{x}_t$ from a convex set $\mathcal{K} \subseteq \mathbb{R}^n$. Then, the adversary chooses a convex function $f_t(\cdot) : \mathbb{R}^n \mapsto \mathbb{R}$, which causes the player a loss $f_t(\mathbf{x}_t)$ but is revealed at the end of round $t + d_t - 1$, where $d_t \geq 1$ denotes an arbitrary delay. The goal of the player is to minimize the regret $\text{Reg}(T) = \sum_{t=1}^{T} f_t(\mathbf{x}_t) - \min_{\mathbf{x} \in \mathcal{K}} \sum_{t=1}^{T} f_t(\mathbf{x})$, i.e., the gap between the cumulative loss of the player and that of an optimal fixed action, where $T$ is the number of total rounds.

Over the past decades, plenty of algorithms and theoretical guarantees have been proposed for this problem [Weinberger and Ordentlich, 2002, Langford et al., 2009, Joulani et al., 2013, Quanrud and Khashabi, 2015, Joulani et al., 2016, Héliou et al., 2020, Flaspohler et al., 2021, Wan et al., 2022a,b, Bistritz et al., 2022]. However, the vast majority of them assume that the full information or gradients

of delayed functions are available for updating the action, which is not necessarily satisfied in reality. For example, in online routing [Awerbuch and Kleinberg, 2008], the player selects a path through a given network for some packet, and its loss is measured by the time length of the path. Although this loss value can be observed after the packet arrives at the destination, the player rarely has access to the congestion pattern of the entire network [Hazan, 2016]. To address this limitation, it is natural to investigate a more challenging setting, namely bandit convex optimization (BCO) with delayed feedback, where only the loss value $f_t(\mathbf{x}_t)$ is revealed at the end of round $t + d_t - 1$.

It is well known that in the non-delayed BCO, bandit gradient descent (BGD), which performs the gradient descent step based on a one-point estimator of the gradient, enjoys a regret bound of $O(\sqrt{n}T^{3/4})$ [Flaxman et al., 2005]. Despite its simplicity, without additional assumptions on functions, there does not exist any practical algorithm that can improve the regret of BGD. Therefore, a few studies have proposed to extend BGD and its regret bound into the delayed setting [Héliou et al., 2020, Bistritz et al., 2022]. Specifically, Héliou et al. [2020] first propose an algorithm called gradient-free online learning with delayed feedback (GOLD), which utilizes the oldest received but not utilized loss value to perform an update similar to BGD at each round. Let $d = \max\{d_1, \ldots, d_T\}$ denote the maximum delay. According to the analysis of Héliou et al. [2020], GOLD can achieve a regret bound of $O(\sqrt{n}T^{3/4} + (nd)^{1/3}T^{2/3})$, which matches the $O(\sqrt{n}T^{3/4})$ regret of BGD in the non-delayed setting for $d = O(\sqrt{n}T^{1/4})$. Very recently, Bistritz et al. [2022] develop an improved variant of GOLD by utilizing all received but not utilized loss values one by one at each round, and reduce the regret bound to $O(\sqrt{n}T^{3/4} + (n\bar{d})^{1/3}T^{2/3})$,[1] where $\bar{d} = (1/T)\sum_{t=1}^{T} d_t$ is the average delay. However, there still exists a large gap between the delay-dependent part in the improved bound and an existing $\Omega(\sqrt{dT})$ lower bound [Bistritz et al., 2022]. It remains unclear whether this gap can be filled, especially by improving the existing upper bound.

In this paper, we provide an affirmative answer to this question in the worst case, where $\bar{d}$ is very close to $d$. Specifically, we first develop a new algorithm, namely delayed follow-the-bandit-leader (D-FTBL), and show that it enjoys a regret bound of $O(\sqrt{n}T^{3/4} + \sqrt{dT})$ in general. Notice that both the $O((nd)^{1/3}T^{2/3})$ and $O((n\bar{d})^{1/3}T^{2/3})$ terms in previous regret bounds [Héliou et al., 2020, Bistritz et al., 2022] can be attributed to the joint effect of the delays, and the one-point gradient estimator, especially its large variance depending on the exploration radius. To improve the regret, besides the one-point gradient estimator, we further incorporate the delayed bandit feedback with a blocking update mechanism, i.e., dividing total $T$ rounds into several equally-sized blocks and only updating the action at the end of each block. Despite its simplicity, there exist two nice properties about the cumulative estimated gradients at each block.

- First, with an appropriate block size, its variance becomes proportional to only the block size without extra dependence on the exploration radius.
- Second, the block-level delay, i.e., the number of blocks waiting for computing the cumulative estimated gradients at each block, is in reverse proportion to the block size.

Surprisingly, by combining these properties, the previous joint effect of the delays and the one-point gradient estimator can be decoupled, which is critical for deriving our regret bound. Compared with the existing results, in the worst case, our regret bound matches the $O(\sqrt{n}T^{3/4})$ regret of the non-delayed BGD for a larger amount of delays, i.e., $d = O(n\sqrt{T})$, and the delay-dependent part, i.e., $O(\sqrt{dT})$, matches the lower bound. Moreover, it is worth noting that our regret bound actually is better than that of Bistritz et al. [2022] as long as $d$ is not larger than $O((n\bar{d})^{2/3}T^{1/3})$, which even covers the case with $\bar{d} = O(1)$ partially. To the best of our knowledge, this is the first work that shows the benefit of the blocking update mechanism in delayed BCO, though it is commonly utilized to develop projection-free algorithms for efficiently dealing with complicated action sets [Zhang et al., 2019, Garber and Kretzu, 2020, Hazan and Minasyan, 2020, Wan et al., 2020, 2022c, Wang et al., 2023, 2024b].

Furthermore, we consider the special case of delayed BCO with strongly convex functions. In the non-delayed setting, Agarwal et al. [2010] have shown that BGD can improve the regret from $O(\sqrt{n}T^{3/4})$ to $O((nT)^{2/3}\log^{1/3}T)$ by exploiting the strong convexity. If functions are also smooth and the action set is unconstrained, BGD has been extended to achieve an $O(n\sqrt{T\log T})$ regret

---

[1]Note that Bistritz et al. [2022] actually only argue a regret bound of $O(nT^{3/4} + \sqrt{n}\bar{d}^{1/3}T^{2/3})$. However, as discussed in our Appendix F, it is not hard to derive this refined bound by tuning parameters more carefully.

bound [Agarwal et al., 2010]. Analogous to these improvements, we prove that our D-FTBL can achieve a regret bound of $O((nT)^{2/3} \log^{1/3} T + d \log T)$ for strongly convex functions, and its simple extension enjoys a regret bound of $O(n\sqrt{T \log T} + d \log T)$ for strongly convex and smooth functions over unconstrained action sets. These regret bounds also match those of BGD in the non-delayed setting for a relatively large amount of delay. Moreover, the $O(d \log T)$ part in these two bounds matches an $\Omega(d \log T)$ lower bound adapted from the easier full-information setting with strongly convex and smooth functions [Weinberger and Ordentlich, 2002].

## 2 Related work

In this section, we briefly review the related work on online convex optimization (OCO) and bandit convex optimization (BCO), as well as delayed feedback.

### 2.1 Standard OCO and BCO

If $d_t = 1$ for all $t \in [T]$, OCO with delayed feedback reduces to the standard OCO [Zinkevich, 2003]. Online gradient descent (OGD) [Zinkevich, 2003, Hazan et al., 2007] is one of the most popular algorithm for this problem, which simply updates the action $\mathbf{x}_t$ via a gradient descent step based on $\nabla f_t(\mathbf{x}_t)$. By using appropriate step sizes, OGD can achieve $O(\sqrt{T})$ and $O(\log T)$ regret bounds for convex and strongly convex functions, respectively. Follow-the-regularized-leader (FTRL) [Hazan et al., 2007, Shalev-Shwartz, 2011, Hazan, 2016] is an alternative algorithm, which chooses the new action by minimizing the linear approximation of cumulative loss functions under some regularization. With appropriate regularization, FTRL achieves the same $O(\sqrt{T})$ and $O(\log T)$ regret bounds as OGD. Moreover, Abernethy et al. [2008] have presented a lower bound of $\Omega(\sqrt{T})$ for convex functions, and a refined lower bound of $\Omega(\log T)$ for strongly convex functions, which implies that both OGD and FTRL are optimal.

BCO is a special yet more challenging case of OCO, where the player can only receive the loss value $f_t(\mathbf{x}_t)$ at each round $t$. The first algorithm for BCO is bandit gradient descent (BGD) [Flaxman et al., 2005], which replaces the exact gradient used in OGD with an estimated gradient based on the single loss value (known as the classical one-point gradient estimator). By incorporating the approximation error of gradients into the regret analysis of OGD, Flaxman et al. [2005] establish an $O(\sqrt{n}T^{3/4})$ regret bound for BGD with convex functions. Later, Agarwal et al. [2010] show that BGD enjoys an $O((nT)^{2/3} \log^{1/3} T)$ regret bound for strongly convex functions, and can be extended to achieve an $O(n\sqrt{T \log T})$ regret bound in the special case of unconstrained BCO with strongly convex and smooth functions. Saha and Tewari [2011] develop a new algorithm for BCO with smooth functions, and establish the $O((nT)^{2/3} \log^{1/3} T)$ regret bound without the strongly convex assumption. van der Hoeven et al. [2020] propose novel BCO algorithms, which adaptively improve the previous regret bounds for convex and smooth functions if the norm of the comparator is small. By revisiting the case with strongly convex and smooth functions, several algorithms [Hazan and Levy, 2014, Ito, 2020] have been developed to achieve the $O(n\sqrt{T \log T})$ regret bound in the constrained setting.

Moreover, a series of studies [Bubeck and Eldan, 2016, Hazan and Li, 2016, Bubeck et al., 2017, Lattimore, 2020, Bubeck et al., 2021] have been devoted to designing nearly optimal algorithms, which almost match the $\Omega(n\sqrt{T})$ lower bound for the general BCO [Shamir, 2013] without any additional assumption. However, the running time of their algorithms are either exponential in $n$ and $T$, or polynomial with a high degree on $n$ and $T$, which is not suitable for practical large-scale applications. We refer the interested reader to Lattimore [2024] for a comprehensive survey on BCO. Additionally, we notice that BCO is closely related to the zero-order stochastic optimization (ZOSO) problem [Duchi et al., 2015, Bach and Perchet, 2016, Shamir, 2017], where the stochastic values are available for minimizing a fixed loss function. However, ZOSO is less challenging than BCO in the sense that it does not need to deal with time-varying functions and is usually allowed to query the loss value at two points per iteration.

### 2.2 OCO and BCO with delays

The seminal work of Weinberger and Ordentlich [2002] first considers the case with a fixed delay, i.e., $d_t = d$ for all $t \in [T]$, and proposes a black-box technique that can covert any traditional OCO

algorithm into the delayed setting. The main idea is to maintain $d$ instances of the traditional algorithm, and alternately utilize these instances to generate the new action. If the regret of the traditional algorithm is bounded by $\text{Reg}(T)$, this technique can achieve an $d\,\text{Reg}(T/d)$ regret bound. Moreover, there exist $\Omega(\sqrt{dT})$ and $\Omega(d\log T)$ lower bounds for convex functions, and strongly convex and smooth functions, respectively [Weinberger and Ordentlich, 2002]. However, the delays are not always fixed in practice, and its space complexity is $d$ times as much as that of the traditional algorithm, which could be prohibitively resource-intensive. Although Joulani et al. [2013] have generalized this technique to deal with arbitrary delays, the space complexity remains high. Besides these black-box techniques, there exists a surge of interest in developing and analyzing specialized algorithms for delayed OCO [Langford et al., 2009, McMahan and Streeter, 2014, Quanrud and Khashabi, 2015, Joulani et al., 2016, Li et al., 2019, Flaspohler et al., 2021, Wan et al., 2022a,b, 2023, 2024], which do not require additional computational resources.

Despite the great flourish of research on OCO with delays and BCO, delayed BCO has rarely been investigated. GOLD [Héliou et al., 2020] is the first algorithm for this problem, which originally has the $O(\sqrt{n}T^{3/4} + (nd)^{1/3}T^{2/3})$ regret, and is further refined to enjoy the $O(\sqrt{n}T^{3/4} + (n\bar{d})^{1/3}T^{2/3})$ regret [Bistritz et al., 2022]. However, Bistritz et al. [2022] also present an unmatched lower bound of $\Omega(\sqrt{dT})$. Although two recent advances in a more complicated bandit non-stochastic control problem [Gradu et al., 2020, Sun et al., 2023] provide some intermediate results about OGD and FTRL with the delayed bandit feedback, they focus on the case with a fixed delay and can only recover the $O(\sqrt{n}T^{3/4} + (nd)^{1/3}T^{2/3})$ regret in general. In this paper, we take one further step toward understanding the effect of arbitrary delays on BCO by establishing improved regret bounds such that the delay-independent part is equal to the regret of BGD, and the delay-dependent part matches the lower bound in the worst case. Moreover, we notice that although the block-box technique of Joulani et al. [2013] can also convert BGD into the delayed setting, it only achieves an $O(\sqrt{n}d^{1/4}T^{3/4})$ regret bound for convex functions, which is much worse than that of GOLD and our algorithm.

## 3    Main results

In this section, we first introduce the necessary preliminaries including definitions, assumptions, and an algorithmic ingredient. Then, we present our improved algorithm for BCO with delayed feedback, as well as the corresponding theoretical guarantees.

### 3.1    Preliminaries

We first recall two standard definitions about the smoothness and strong convexity of functions [Boyd and Vandenberghe, 2004].

**Definition 1.** *A function $f(\mathbf{x}) : \mathbb{R}^n \to \mathbb{R}$ is called $\beta$-smooth over $\mathcal{K}$ if for all $\mathbf{x}, \mathbf{y} \in \mathcal{K}$, it holds that $f(\mathbf{y}) \leq f(\mathbf{x}) + \langle \nabla f(\mathbf{x}), \mathbf{y} - \mathbf{x} \rangle + \frac{\beta}{2}\|\mathbf{y} - \mathbf{x}\|_2^2$.*

**Definition 2.** *A function $f(\mathbf{x}) : \mathbb{R}^n \to \mathbb{R}$ is called $\alpha$-strongly convex over $\mathcal{K}$ if for all $\mathbf{x}, \mathbf{y} \in \mathcal{K}$, it holds that $f(\mathbf{y}) \geq f(\mathbf{x}) + \langle \nabla f(\mathbf{x}), \mathbf{y} - \mathbf{x} \rangle + \frac{\alpha}{2}\|\mathbf{y} - \mathbf{x}\|_2^2$.*

Note that as proved by Hazan and Kale [2012], any $\alpha$-strongly convex function $f(\mathbf{x}) : \mathbb{R}^n \mapsto \mathbb{R}$ over the convex set $\mathcal{K}$ ensures that

$$\frac{\alpha}{2}\|\mathbf{x} - \mathbf{x}^*\|_2^2 \leq f(\mathbf{x}) - f(\mathbf{x}^*) \tag{1}$$

for any $\mathbf{x} \in \mathcal{K}$, where $\mathbf{x}^* = \arg\min_{\mathbf{x} \in \mathcal{K}} f(\mathbf{x})$.

Then, following previous studies on BCO [Flaxman et al., 2005, Héliou et al., 2020, Garber and Kretzu, 2020, 2021], we introduce some common assumptions.

**Assumption 1.** *The convex set $\mathcal{K}$ is full-dimensional and contains the origin, and there exist two constants $r, R > 0$ such that $r\mathcal{B}^n \subseteq \mathcal{K} \subseteq R\mathcal{B}^n$, where $\mathcal{B}^n$ denotes the unit Euclidean ball centered at the origin in $\mathbb{R}^n$.*

**Assumption 2.** *All loss functions are $G$-Lipschitz over $\mathcal{K}$, i.e., for all $\mathbf{x}, \mathbf{y} \in \mathcal{K}$ and $t \in [T]$, it holds that $|f_t(\mathbf{x}) - f_t(\mathbf{y})| \leq G\|\mathbf{x} - \mathbf{y}\|_2$.*

**Assumption 3.** *The absolute value of all loss functions over $\mathcal{K}$ are bounded by $M$, i.e., for all $\mathbf{x} \in \mathcal{K}$ and $t \in [T]$, it holds that $|f_t(\mathbf{x})| \leq M$. Additionally, all loss functions are chosen beforehand, i.e., the adversary is oblivious.*

Finally, we introduce the one-point gradient estimator [Flaxman et al., 2005], which is a standard technique for exploiting the bandit feedback. Given a function $f(\mathbf{x}) : \mathbb{R}^n \mapsto \mathbb{R}$, we can define the $\delta$-smoothed version of $f(\mathbf{x})$ as

$$\hat{f}_\delta(\mathbf{x}) = \mathbb{E}_{\mathbf{u} \sim \mathcal{B}^n}[f(\mathbf{x} + \delta \mathbf{u})] \tag{2}$$

where the parameter $\delta \in (0, 1)$ is the so-called exploration radius. As proved by Flaxman et al. [2005], the $\delta$-smoothed version satisfies the following lemma.

**Lemma 1.** *(Lemma 1 in Flaxman et al. [2005]) Given a function $f(\mathbf{x}) : \mathbb{R}^n \mapsto \mathbb{R}$ and a constant $\delta \in (0, 1)$, its $\delta$-smoothed version $\hat{f}_\delta(\mathbf{x})$ defined in* (2) *ensures*

$$\nabla \hat{f}_\delta(\mathbf{x}) = \mathbb{E}_{\mathbf{u} \sim \mathcal{S}^n} \left[ \frac{n}{\delta} f(\mathbf{x} + \delta \mathbf{u}) \mathbf{u} \right]$$

*where $\mathcal{S}^n$ denotes the unit Euclidean sphere centered at the origin in $\mathbb{R}^n$.*

From Lemma 1, the randomized vector $\frac{n}{\delta} f(\mathbf{x} + \delta \mathbf{u}) \mathbf{u}$, which can be computed by only utilizing a single loss value, is an unbiased estimator of $\nabla \hat{f}_\delta(\mathbf{x})$. Moreover, Flaxman et al. [2005] have also shown that $\hat{f}_\delta(\mathbf{x})$ is close to the original function $f(\mathbf{x})$ over a shrunk set

$$\mathcal{K}_\delta = (1 - \delta/r)\mathcal{K} = \{(1 - \delta/r)\mathbf{x} | \mathbf{x} \in \mathcal{K}\}. \tag{3}$$

Therefore, this one-point estimator can be utilized as a good substitute for the gradient $\nabla f(\mathbf{x})$ in the bandit setting. For example, we notice that at each round $t$, BGD [Flaxman et al., 2005] first plays an action $\mathbf{x}_t = \mathbf{y}_t + \mathbf{u}_t$, where $\mathbf{y}_t \in \mathcal{K}_\delta$ and $\mathbf{u}_t \sim \mathcal{S}^n$, and then updates $\mathbf{y}_t$ as

$$\mathbf{y}_{t+1} = \Pi_{\mathcal{K}_\delta} \left( \mathbf{y}_t - \frac{\eta_t n}{\delta} f_t(\mathbf{x}_t) \mathbf{u}_t \right) \tag{4}$$

where $\Pi_{\mathcal{K}_\delta}(\mathbf{y}) = \mathrm{argmin}_{\mathbf{x} \in \mathcal{K}_\delta} \|\mathbf{x} - \mathbf{y}\|_2^2$ denotes the projection onto the set $\mathcal{K}_\delta$, and $\eta_t$ is the step size.

### 3.2 Our improved algorithm

Before introducing our algorithm, we first briefly discuss the joint effect of the delays and the bandit feedback in GOLD [Héliou et al., 2020], which will provide insights for our improvements. Recall that in the delayed setting, the loss value $f_t(\mathbf{x}_t)$ will be delayed to the end of round $t + d_t - 1$, and thus the player can only receive $\{f_k(\mathbf{x}_k) | k \in \mathcal{F}_t\}$ at the end of round $t$, where $\mathcal{F}_t = \{k | k + d_k - 1 = t\}$. Since the set $\mathcal{F}_t$ may not contain the round $t$, the vanilla BGD in (4) is no longer valid. To address this issue, GOLD [Héliou et al., 2020] replaces $f_t(\mathbf{x}_t)$ in (4) with the oldest received but not utilized loss value at the end of round $t$. Intuitively, the update of this approach is $O(d)$ rounds slower than that of the vanilla BGD, which is analogous to those delayed OCO algorithms. However, due to the use of the one-point gradient estimator, the slower update causes a difference of $O(\eta d n/\delta)$ between its action and that of BGD, and the cumulative difference will bring additional regret of $O(T\eta d n/\delta)$, where a constant step size $\eta_t = \eta$ is discussed for brevity. Note that from the standard analysis of BGD, to control the total exploration cost, the value of $1/\delta$ should be sublinear in $T$. Therefore, it will amplify the effect of delays, and finally results in the $O(\sqrt{n}T^{3/4} + (nd)^{1/3}T^{2/3})$ regret [Héliou et al., 2020].

To reduce the effect of delays, we propose to incorporate the delayed bandit feedback with a blocking update mechanism [Zhang et al., 2019, Garber and Kretzu, 2020]. Specifically, we divide the total $T$ rounds into $T/K$ blocks, each with $K$ rounds, where $T/K$ is assumed to be an integer without loss of generality. For each block $m \in [T/K]$, we only maintain a preparatory action $\mathbf{y}_m \in \mathcal{K}_\delta$, and play $\mathbf{x}_t = \mathbf{y}_m + \delta \mathbf{u}_t$ with $\mathbf{u}_t \sim \mathcal{S}^n$ at each round $t$ in the block. Due to the randomness of $\mathbf{u}_t$ and the independence of $\mathbf{x}_t$ in the same block, it is not hard to verify that for each block $m \in [T/K]$, the sum of randomized gradients generated by the one-point estimator, i.e., $\nabla_m = \sum_{t=(m-1)K+1}^{mK} \frac{n}{\delta} f_t(\mathbf{x}_t) \mathbf{u}_t$, satisfies (see Lemma 5 presented in Section 3.4 for details)

$$\mathbb{E}[\|\nabla_m\|_2] = O(\sqrt{Kn^2/\delta^2} + K).$$

By using an appropriate block size of $K = O(n^2/\delta^2)$, this upper bound will be $\mathbb{E}[\|\nabla_m\|_2] = O(K)$. By contrast, without the blocking update mechanism, one can only achieve $\mathbb{E}[\|\nabla_m\|_2] = O(Kn/\delta)$. Moreover, we notice that the cumulative estimated gradients $\nabla_m$ will be delayed at most $O(d/K)$

---

**Algorithm 1** Delayed Follow-The-Bandit-Leader

1: **Input:** $\delta, K, \alpha$, and $\eta > 0$ if $\alpha = 0$
2: **Initialization:** set $\bar{\mathbf{g}}_0 = \mathbf{0}$ and choose $\mathbf{y}_1 \in \mathcal{K}_\delta$ arbitrarily
3: **for** $m = 1, 2, \ldots, T/K$ **do**
4:     **for** $t = (m-1)K + 1, \ldots, mK$ **do**
5:         Play $\mathbf{x}_t = \mathbf{y}_m + \delta \mathbf{u}_t$, where $\mathbf{u}_t \sim \mathcal{S}^n$
6:         Query $f_t(\mathbf{x}_t)$, and receive $\{f_k(\mathbf{x}_k) | k \in \mathcal{F}_t\}$
7:         Update $\bar{\mathbf{g}}_t = \bar{\mathbf{g}}_{t-1} + \sum_{k \in \mathcal{F}_t} \frac{n}{\delta} f_k(\mathbf{x}_k) \mathbf{u}_k$
8:     **end for**
9:     Set $\mathcal{R}_m(\mathbf{x}) = \begin{cases} \frac{1}{\eta} \|\mathbf{x} - \mathbf{y}_1\|_2^2 & \text{if } \alpha = 0 \\ \sum_{i=1}^m \frac{K\alpha}{2} \|\mathbf{x} - \mathbf{y}_i\|_2^2 & \text{otherwise} \end{cases}$
10:     $\mathbf{y}_{m+1} = \operatorname{argmin}_{\mathbf{x} \in \mathcal{K}_\delta} \{\langle \bar{\mathbf{g}}_{mK}, \mathbf{x} \rangle + \mathcal{R}_m(\mathbf{x})\}$
11: **end for**

---

blocks, because even the last component $\frac{n}{\delta} f_{mK}(\mathbf{x}_{mK}) \mathbf{u}_{mK}$ is available at the end of round $mK + d - 1$.

As a result, one possible approach to determine $\mathbf{y}_m$ for each block is to extend the update rule of GOLD [Héliou et al., 2020] into the block level with $K = O(n^2/\delta^2)$. Combining with previous discussions, it will reduce the effect of delays on the regret from $O(T\eta dn/\delta)$ to

$$O\left(T\eta \frac{d}{K}\left(\sqrt{\frac{Kn^2}{\delta^2}} + K\right)\right) = O(\eta dT)$$

which is good enough for deriving our desired regret bounds. However, it requires a bit complicated procedure to maintain the cumulative estimated gradients for any block that has not been utilized to update the action. For this reason, instead of utilizing this approach, we incorporate FTRL [Hazan et al., 2007, Hazan, 2016] with the delayed bandit feedback and blocking update mechanism, which provides a more elegant way to utilize the delayed information.

Specifically, we initialize $\mathbf{y}_1 \in \mathcal{K}_\delta$ arbitrarily, and use a variable $\bar{\mathbf{g}}_t$ to record the sum of gradients estimated from all received loss values, i.e., $\bar{\mathbf{g}}_t = \sum_{i=1}^t \sum_{k \in \mathcal{F}_i} \frac{n}{\delta} f_k(\mathbf{x}_k) \mathbf{u}_k$. Then, according to FTRL, an ideal action should be selected by minimizing the linear approximation of cumulative loss functions under some regularization, i.e.,

$$\mathbf{y}_{m+1}^* = \operatorname*{argmin}_{\mathbf{x} \in \mathcal{K}_\delta} \left\{ \sum_{i=1}^m \langle \nabla_i, \mathbf{x} \rangle + \mathcal{R}_m(\mathbf{x}) \right\} \tag{5}$$

where the regularization is set as $\mathcal{R}_m(\mathbf{x}) = \frac{1}{\eta} \|\mathbf{x} - \mathbf{y}_1\|_2^2$ for convex functions [Hazan, 2016] and $\mathcal{R}_m(\mathbf{x}) = \sum_{i=1}^m \frac{K\alpha}{2} \|\mathbf{x} - \mathbf{y}_i\|_2^2$ for $\alpha$-strongly convex functions [Hazan et al., 2007]. Unfortunately, due to the effect of delays, the value of $\sum_{i=1}^m \nabla_i$ required by (5) may not be available. To address this limitation, we generate $\mathbf{y}_{m+1}$ by replacing this term with the sum of all available estimated gradients, i.e., $\bar{\mathbf{g}}_{mK}$.[2]

The detailed procedures are outlined in Algorithm 1, where the input $\alpha$ is the modules of the strong convexity of functions, and it is called delayed follow-the-bandit-leader (D-FTBL).

### 3.3 Theoretical guarantees

We first present the regret bound of our D-FTBL for convex functions.

**Theorem 1.** *Under Assumptions 1, 2, and 3, Algorithm 1 with $\alpha = 0$ ensures*

$$\mathbb{E}[\text{Reg}(T)] \leq \underbrace{\frac{4R^2}{\eta} + \frac{\eta T\gamma}{2K}}_{:=A} + \underbrace{\frac{\eta TG}{2}\sqrt{2\left(\frac{d^2}{K^2} + 4\right)\gamma}}_{:=B} + \underbrace{3\delta GT + \frac{\delta GRT}{r}}_{:=C} \tag{6}$$

*where $\gamma = K\left(\frac{nM}{\delta}\right)^2 + K^2 G^2$.*

---

[2] From the above discussions, one may replace $\sum_{i=1}^m \nabla_i$ with the sum of all available $\nabla_i$. However, we find that simply utilizing $\bar{\mathbf{g}}_{mK}$ can attain the same regret, though they have a slight difference.

**Remark.** To help understanding the regret bound in (6), we notice that the term $A$ actually is derived from the expected regret of the ideal action $\mathbf{y}_m^*$ on a sequence of surrogate losses, and the term $B$ is caused by the cumulative distance between our preparatory action $\mathbf{y}_m$ and the ideal one. Additionally, the term $C$ is caused by the exploration error of the one-point gradient estimator. At first glance, it seems that term $B$ suffers a multiplicative joint effect of the maximum delay $d$ and the exploration radius $\delta$ due to the existence of $\gamma$. However, as discussed before, this joint effect can be decoupled by setting an appropriate block size of $K = O(n^2/\delta^2)$, which allows us to derive an improved regret bound. Specifically, by substituting $\alpha = 0$, $K = n\sqrt{T}$, $\eta = 1/\max\{\sqrt{Td}, \sqrt{n}T^{3/4}\}$, and $\delta = c\sqrt{n}T^{-1/4}$ into (6), where $c$ is a constant such that $\delta < r$, our D-FTBL can enjoy

$$\mathbb{E}\left[\mathrm{Reg}(T)\right] \leq O\left(\sqrt{n}T^{3/4} + \sqrt{dT}\right) \tag{7}$$

for convex functions.[3] It is tighter than the $O(\sqrt{n}T^{3/4} + (nd)^{1/3}T^{2/3})$ regret of GOLD [Héliou et al., 2020], and matches the $O(\sqrt{n}T^{3/4})$ regret bound of BGD in the non-delayed setting as long as $d$ is not larger than $O(n\sqrt{T})$. Even for $d = \Omega(n\sqrt{T})$, our regret bound is dominated by the $O(\sqrt{dT})$ part, which matches the $\Omega(\sqrt{dT})$ lower bound [Bistritz et al., 2022] in the worst case. Moreover, although the $O(\sqrt{n}T^{3/4} + (n\bar{d})^{1/3}T^{2/3})$ regret bound of Bistritz et al. [2022] could benefit from a small average delay, it is also worse than our regret bound when $d$ is not larger than $O((n\bar{d})^{2/3}T^{1/3})$.

Then, we establish an improved regret bound for $\alpha$-strongly convex functions.

**Theorem 2.** *Under Assumptions 1, 2, and 3, if all functions are $\alpha$-strongly convex, Algorithm 1 with $\alpha > 0$ ensures*

$$\mathbb{E}\left[\mathrm{Reg}(T)\right] \leq \underbrace{\frac{2\gamma C_T}{\alpha K} + C_T' R\sqrt{\gamma}}_{:=A'} + \underbrace{\frac{GC_T}{\alpha}\sqrt{2\left(\frac{d^2}{K^2} + 4\right)\gamma}}_{:=B'} + 3\delta GT + \frac{\delta GRT}{r} \tag{8}$$

*where $\gamma = K\left(\frac{nM}{\delta}\right)^2 + K^2 G^2$, $C_T = 1 + \ln T$, and $C_T' = 6 + 4\ln T$.*

**Remark.** By comparing Theorem 2 with Theorem 1, we find that the strongly convexity can be exploited to reduce the expected regret of the ideal action $\mathbf{y}_m^*$ on the surrogate losses, and the cumulative distance between our preparatory action $\mathbf{y}_m$ and the ideal one, i.e., improving terms $A$ and $B$ in (6) to terms $A'$ and $B'$ in (8). By further substituting $\alpha > 0$, $K = (nT)^{2/3}\ln^{-2/3}T$, and $\delta = cn^{2/3}T^{-1/3}\ln^{1/3}T$ into (8), where $c$ is a constant such that $\delta < r$, our D-FTBL can enjoy

$$\mathbb{E}\left[\mathrm{Reg}(T)\right] \leq O\left((nT)^{2/3}\log^{1/3}T + d\log T\right) \tag{9}$$

for strongly convex functions. This regret bound is tighter than the above $O(\sqrt{n}T^{3/4} + \sqrt{dT})$ regret bound achieved by only utilizing the convexity condition, and can match the $O((nT)^{2/3}\log^{1/3}T)$ regret bound of BGD in the non-delayed setting as long as $d$ is not larger than $O((nT/\log T)^{2/3})$. Even if $d = \Omega((nT/\log T)^{2/3})$, it is dominated by the $O(d\log T)$ part, which matches the $\Omega(d\log T)$ lower bound [Weinberger and Ordentlich, 2002], and thus cannot be improved. Moreover, different from the case with convex functions, the parameters for achieving the bound in (9) do not require the information of delays.

Furthermore, we consider the unconstrained case, i.e., $\mathcal{K} = \mathbb{R}^n$, with $\alpha$-strongly convex and $\beta$-smooth functions, and extend our D-FTBL to achieve a better regret bound. Specifically, without the boundedness of $\mathcal{K}$, Assumptions 2 and 3 may no longer hold over the entire space [Agarwal et al., 2010]. Therefore, we first introduce a weaker assumption on the Lipschitz continuity, i.e, all loss functions are $G$-Lipschitz at $\mathbf{0}$. Combining with (1), it is not hard to verify that the fixed optimal action $\mathbf{x}^* = \mathrm{argmin}_{\mathbf{x} \in \mathbb{R}^n} \sum_{t=1}^T f_t(\mathbf{x})$ satisfies

$$\|\mathbf{x}^*\|_2 \leq \frac{2G}{\alpha}. \tag{10}$$

---

[3]One may notice that the step size for achieving this result depends on the maximum delay $d$, which may be unknown beforehand. Fortunately, as discussed in previous studies [Quanrud and Khashabi, 2015, Wan et al., 2024], there exists a standard solution—utilizing the "doubling trick" [Cesa-Bianchi et al., 1997] to adaptively estimate the maximum delay $d$ and adjust the step size, which can attain the same bound as in (7).

As a result, the player only needs to select actions from the following set

$$\mathcal{K}' = \left\{ \mathbf{x} \in \mathbb{R}^n \,\middle|\, \|\mathbf{x}\|_2 \le \frac{2G}{\alpha} \right\} \tag{11}$$

which satisfies Assumption 1 with $r = R = 2G/\alpha$, and it is natural to further assume that all loss functions satisfy Assumptions 2 and 3 over the set $\mathcal{K}'$. Now, we can apply our D-FTBL over the shrink set of $\mathcal{K}'$, i.e.,

$$\mathcal{K}'_\delta = (1 - \delta/r)\mathcal{K}' = \left(1 - \frac{\alpha\delta}{2G}\right)\mathcal{K}' \tag{12}$$

instead of the original $\mathcal{K}_\delta$, and establish the following regret bound.

**Theorem 3.** *Let $\mathcal{K} = \mathbb{R}^n$. If all loss functions are $\alpha$-strongly convex and $\beta$-smooth over $\mathcal{K}$, and Assumptions 2 and 3 hold over $\mathcal{K}'$ defined in* (11)*, applying Algorithm 1 with $\alpha > 0$ over $\mathcal{K}'_\delta$ defined in* (12) *ensures*

$$\mathbb{E}\left[\text{Reg}(T)\right] \le \frac{2\gamma C_T}{\alpha K} + \frac{2C'_T G\sqrt{\gamma}}{\alpha} + \frac{GC_T}{\alpha}\sqrt{2\left(\frac{d^2}{K^2} + 4\right)\gamma} + \underbrace{\beta\delta^2 T + \frac{\beta\delta^2 GT}{\alpha}}_{:=C'} \tag{13}$$

*where $\gamma = K\left(\frac{nM}{\delta}\right)^2 + K^2 G^2$, $C_T = 1 + \ln T$, and $C'_T = 6 + 4\ln T$.*

**Remark.** By comparing Theorem 3 with Theorem 2, we find that the exploration error of the one-point gradient estimator is reduced, i.e., improving the last two terms in (8) to the term $C'$ in (13). Then, by substituting $\alpha > 0$, $K = n\sqrt{T/\ln T}$, and $\delta = cn^{1/2}T^{-1/4}\ln^{1/4}T$ into (13), where $c$ is a constant such that $\delta < 2G/\alpha$, we can achieve an $O\left(n\sqrt{T\log T} + d\log T\right)$ regret bound for strongly convex and smooth functions in the unconstrained case. It is better than the $O((nT)^{2/3}\log^{1/3}T + d\log T)$ regret bound achieved by only utilizing the strong convexity. Moreover, this bound matches the $O(n\sqrt{T\log T})$ regret bound achieved by using BGD in the non-delayed setting as long as $d$ is not larger than $O(n\sqrt{T/\log T})$. Otherwise, it is dominated by the $O(d\log T)$ part, which cannot be improved as discussed before.

### 3.4 Analysis: proof of Theorem 1

Due to the limitation of space, here we only prove Theorem 1, and the omitted proofs can be found in the appendix. Specifically, let $\tilde{\mathbf{x}}^* = (1 - \delta/r)\mathbf{x}^*$ where $\mathbf{x}^* \in \text{argmin}_{\mathbf{x}\in\mathcal{K}}\sum_{t=1}^T f_t(\mathbf{x})$, and recall the ideal action defined in (5). As in Lemma 2, we first notice that the expected regret of Algorithm 1 can be bounded by the sum of three parts including the expected regret of ideal actions on some surrogate losses, the cumulative distance between $\mathbf{y}_m$ and the ideal one, and the exploration error of the one-point gradient estimator.

**Lemma 2.** *Under Assumptions 1 and 2, Algorithm 1 with $\alpha = 0$ ensures*

$$\mathbb{E}\left[\text{Reg}(T)\right] \le \mathbb{E}\left[\sum_{m=1}^{T/K}\langle\nabla_m, \mathbf{y}_m^* - \tilde{\mathbf{x}}^*\rangle + KG\sum_{m=1}^{T/K}\|\mathbf{y}_m - \mathbf{y}_m^*\|_2\right] + 3\delta GT + \frac{\delta GRT}{r}. \tag{14}$$

Note that the part regarding the exploration error in (14) is exactly the same as the term $C$ in (6). So, we only need to analyze the first two parts in (14). For the first part, we define surrogate losses as $\ell_1(\mathbf{x}) = \langle\nabla_1, \mathbf{x}\rangle + \frac{1}{\eta}\|\mathbf{x} - \mathbf{y}_1\|_2$ and $\ell_m(\mathbf{x}) = \langle\nabla_m, \mathbf{x}\rangle$ for any $m = 2, \ldots, T/K$. Combining with (5) for convex functions, it is easy to verify that $\mathbf{y}_{m+1}^* = \text{argmin}_{\mathbf{x}\in\mathcal{K}_\delta}\sum_{i=1}^m \ell_i(\mathbf{x})$. Then, we introduce the following lemma to bound the regret of $\mathbf{y}_2^*, \ldots, \mathbf{y}_{T/K+1}^*$ on $\ell_1(\cdot), \ldots, \ell_{T/K}(\cdot)$.

**Lemma 3.** *(Lemma 6.6 in Garber and Hazan [2016]) Let $\{\ell_t(\mathbf{x})\}_{t=1}^T$ be a sequence of functions over a set $\mathcal{K}$, and let $\mathbf{x}_t^* \in \text{argmin}_{\mathbf{x}\in\mathcal{K}}\sum_{i=1}^t \ell_i(\mathbf{x})$ for any $t \in [T]$. Then, it holds that $\sum_{t=1}^T \ell_t(\mathbf{x}_t^*) - \min_{\mathbf{x}\in\mathcal{K}}\sum_{t=1}^T \ell_t(\mathbf{x}) \le 0$.*

Specifically, by applying Lemma 3, we have $\sum_{m=1}^{T/K}\ell_m(\mathbf{y}_{m+1}^*) - \sum_{m=1}^{T/K}\ell_m(\tilde{\mathbf{x}}^*) \le 0$. Combining this inequality with Assumption 1, we have

$$\sum_{m=1}^{T/K}\langle\nabla_m, \mathbf{y}_{m+1}^* - \tilde{\mathbf{x}}^*\rangle \le \frac{\|\tilde{\mathbf{x}}^* - \mathbf{y}_1\|_2^2}{\eta} - \frac{\|\mathbf{y}_2^* - \mathbf{y}_1\|_2^2}{\eta} \le \frac{4R^2}{\eta}. \tag{15}$$

Moreover, to replace $\mathbf{y}_{m+1}^*$ in the left side of (15) with $\mathbf{y}_m^*$, we introduce the following lemma.

**Lemma 4.** *(Lemma 5 in Duchi et al. [2011]) Let $\Pi_{\mathcal{K}}(\mathbf{u}, \eta) = \arg\min_{\mathbf{x} \in \mathcal{K}} \left\{ \langle \mathbf{u}, \mathbf{x} \rangle + \frac{1}{\eta} \|\mathbf{x}\|_2^2 \right\}$. We have $\|\Pi_{\mathcal{K}}(\mathbf{u}, \eta) - \Pi_{\mathcal{K}}(\mathbf{v}, \eta)\|_2 \le \frac{\eta}{2} \|\mathbf{u} - \mathbf{v}\|_2$.*

Combining Lemma 4 with (5) for convex functions, we have

$$\|\mathbf{y}_m^* - \mathbf{y}_{m+1}^*\|_2 \le \frac{\eta}{2} \left\| \left( \sum_{i=1}^{m-1} \nabla_i - \frac{2\mathbf{y}_1}{\eta} \right) - \left( \sum_{i=1}^{m} \nabla_i - \frac{2\mathbf{y}_1}{\eta} \right) \right\|_2 = \frac{\eta}{2} \|\nabla_m\|_2. \tag{16}$$

Then, combining (15) with (16), we have

$$\sum_{m=1}^{T/K} \langle \nabla_m, \mathbf{y}_m^* - \tilde{\mathbf{x}}^* \rangle = \sum_{m=1}^{T/K} \langle \nabla_m, \mathbf{y}_{m+1}^* - \tilde{\mathbf{x}}^* + \mathbf{y}_m^* - \mathbf{y}_{m+1}^* \rangle$$
$$\le \frac{4R^2}{\eta} + \sum_{m=1}^{T/K} \|\nabla_m\|_2 \|\mathbf{y}_m^* - \mathbf{y}_{m+1}^*\|_2 \le \frac{4R^2}{\eta} + \frac{\eta}{2} \sum_{m=1}^{T/K} \|\nabla_m\|_2^2. \tag{17}$$

We notice that the term $\|\nabla_m\|_2^2$ in (17) can directly benefit from the blocking update mechanism, as shown by the upper bound in the following lemma.

**Lemma 5.** *Under Assumptions 2 and 3, for any $m \in [T/K]$, Algorithm 1 ensures $\mathbb{E}[\|\nabla_m\|_2^2] \le K \left( \frac{nM}{\delta} \right)^2 + K^2 G^2$.*

However, to completely bound the right side of (14), we still need to analyze $\|\mathbf{y}_m - \mathbf{y}_m^*\|_2$, which is more complicated due to the effect of delays. Specifically, let

$$\mathcal{U}_m = \{1, \ldots, (m-1)K\} \setminus \cup_{t=1}^{(m-1)K} \mathcal{F}_t \tag{18}$$

be the set consisting of the time stamp of loss values that are queried but still not arrive at the end of round $(m-1)K$. By using Lemma 4 again, we have

$$\|\mathbf{y}_m - \mathbf{y}_m^*\|_2 \le \frac{\eta}{2} \left\| \left( \bar{\mathbf{g}}_{(m-1)K} - \frac{2\mathbf{y}_1}{\eta} \right) - \left( \sum_{i=1}^{m-1} \nabla_i - \frac{2\mathbf{y}_1}{\eta} \right) \right\|_2 = \frac{\eta}{2} \left\| \sum_{t \in \mathcal{U}_m} \frac{n}{\delta} f_t(\mathbf{x}_t) \mathbf{u}_t \right\|_2. \tag{19}$$

Moreover, we establish the following lemma regarding the right side of (19).

**Lemma 6.** *Under Assumptions 2 and 3, for any $m \in [T/K]$, Algorithm 1 ensures*

$$\mathbb{E}\left[ \left\| \sum_{t \in \mathcal{U}_m} \frac{n}{\delta} f_t(\mathbf{x}_t) \mathbf{u}_t \right\|_2^2 \right] \le 2 \left( \frac{d^2}{K^2} + 4 \right) \left( K \left( \frac{nM}{\delta} \right)^2 + K^2 G^2 \right).$$

Combining (14), (17), (19), Lemma 5, Lemma 6, and $\gamma = K \left( \frac{nM}{\delta} \right)^2 + K^2 G^2$, we have

$$\mathbb{E}[\text{Reg}(T)] \le \frac{4R^2}{\eta} + \mathbb{E}\left[ \frac{\eta}{2} \sum_{m=1}^{T/K} \|\nabla_m\|_2^2 \right] + KG \sum_{m=1}^{T/K} \mathbb{E}[\|\mathbf{y}_m - \mathbf{y}_m^*\|_2] + 3\delta GT + \frac{\delta GRT}{r}$$
$$\le \frac{4R^2}{\eta} + \frac{\eta T \gamma}{2K} + \frac{\eta TG}{2} \sqrt{2 \left( \frac{d^2}{K^2} + 4 \right) \gamma} + 3\delta GT + \frac{\delta GRT}{r}.$$

## 4  Experiments

In this section, we compare our D-FTBL against GOLD [Héliou et al., 2020] and improved GOLD [Bistritz et al., 2022] by conducting simulation experiments on two publicly available data sets—ijcnn1 and SUSY from the LIBSVM repository [Chang and Lin, 2011]. All algorithms are implemented with Python, and tested on a laptop with 2.4GHz CPU and 16GB memory.

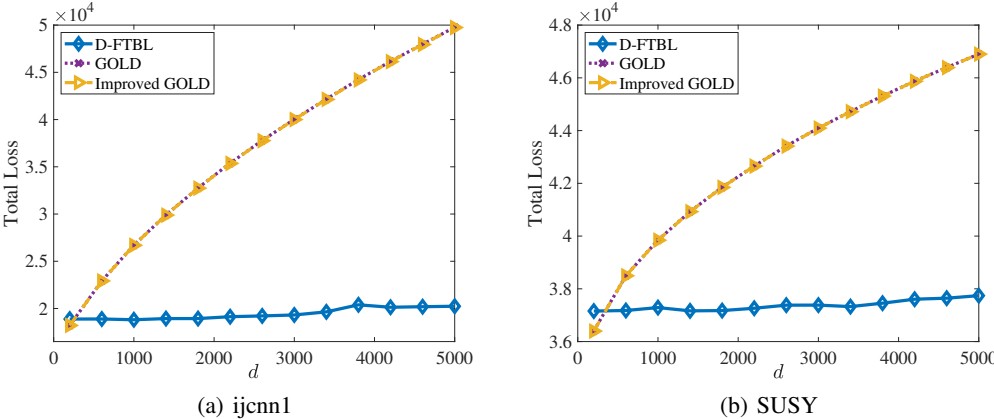

(a) ijcnn1                                                (b) SUSY

Figure 1: Experimental results on delayed online binary classification for ijcnn1 and SUSY.

Specifically, we randomly select $T = 40000$ examples from the original data sets, and consider online binary classification over a convex set $\mathcal{K} = \{\mathbf{x} \in \mathbb{R}^n | \|\mathbf{x}\|_2 \leq 50\}$. The dimensionality of ijcnn1 and SUSY are $n = 22$ and $n = 18$, respectively. In each round $t \in [T]$, the adversary chooses the hinge loss

$$f_t(\mathbf{x}) = \max\left\{1 - y_t \mathbf{w}_t^\top \mathbf{x}, 0\right\}$$

where $\mathbf{w}_t$ and $y_t \in \{-1, 1\}$ are the feature vector and class label of the $t$-th example, respectively. Different values of the maximum delay $d$ in the set $\{200, 600, 1000, \ldots, 5000\}$ have been tried in our experiments. For each specific $d$, to simulate arbitrary delays, $d_t$ is independently and uniformly sampled from $[d]$. In this way, the average delay $\bar{d}$ is equal to $(d+1)/2$ in expectation, and thus is close to the maximum delay.

According to the previous discussions about Theorem 1, we set $\alpha = 0$, $K = \lfloor n\sqrt{T} \rfloor$, $\delta = c\sqrt{n}T^{-1/4}$, and $\eta = c' / \max\{\sqrt{Td}, \sqrt{n}T^{3/4}\}$ for our D-FTBL by tuning these two constants $c$ and $c'$. For those two baselines, we only need to set parameters $\delta$ and $\eta$. In addition to the theoretically suggested value of $\delta$ and $\eta$, we also introduce $c$ and $c'$ as the scale factor, respectively. For all algorithms, $c$ and $c'$ are respectively selected from $\{0.1, 1.0, 10\}$ and $\{0.01, 0.1, \ldots, 100\}$ simply according to their performance for $d = 200$. Moreover, due to the randomness of these algorithms, we repeat them 20 times and report the average of their total loss.

Fig. 1 shows the results of all algorithms on both data sets. We first find that when $d$ increases from 200 to 5000, the total loss of our D-FTBL grows slowly, which is consistent with the dependence of our regret bound on $d$. It is worth noting that $d = 5000$ is larger than $n\sqrt{T}$ in our experiments. Second, from $d = 600$ to $d = 5000$, the total loss of our D-FTBL is better than both GOLD and improved GOLD, which verifies the advantage of our algorithm in the delayed setting. By contrast, due to $\bar{d} \approx d$, the performance of improved GOLD is very close to that of GOLD. Finally, we also notice that D-FTBL is slightly worse than baselines for $d = 200$. However, it is reasonable because the block update mechanism enlarges each delay to be at least the block size, which could result in a slightly larger constant factor in the regret.

## 5   Conclusion and future work

In this paper, we investigate BCO with delayed feedback, and propose a novel algorithm called D-FTBL by exploiting the blocking update mechanism. Our analysis first reveals that it can achieve a regret bound of $O(\sqrt{n}T^{3/4} + \sqrt{dT})$ in general, which improves the delay-dependent part of the existing $O(\sqrt{n}T^{3/4} + (n\bar{d})^{1/3}T^{2/3})$ regret bound as long as $d$ is not larger than $O((n\bar{d})^{2/3}T^{1/3})$. Furthermore, we consider the special case with strongly convex functions, and prove that the regret of D-FTBL can be reduced to $O((nT)^{2/3}\log^{1/3}T + d\log T)$. Finally, if the action sets are unconstrained, we show that D-FTBL can be simply extended to enjoy the $O(n\sqrt{T\log T} + d\log T)$ regret for strongly convex and smooth functions. Nonetheless, there still exist several open problems, which are discussed in the appendix due to the limitation of space.

## Acknowledgments

This work was partially supported by the National Natural Science Foundation of China (62306275, U23A20382), the Zhejiang Province High-Level Talents Special Support Program "Leading Talent of Technological Innovation of TenThousands Talents Program" (No. 2022R52046), the Key Research and Development Program of Zhejiang Province (No. 2023C03192), and the Open Research Fund of the State Key Laboratory of Blockchain and Data Security, Zhejiang University. The authors would also like to thank Chenxu Zhang for helping conduct experiments.

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

## A    Detailed discussions on future work

First, we notice that all our regret bounds depend on the maximum delay, and thus it is natural to investigate whether these bounds can be further improved to be depending on the average delay. It seems highly non-trivial to obtain such results with our D-FTBL because the blocking update mechanism actually enlarges each delay to be at least the block size.

Second, it is appealing to apply our algorithm to an emerging task—memory-efficient fine-tuning of large language models [Malladi et al., 2023, Zhang et al., 2024]. The key insight is that our algorithm only requires the delayed loss value to update the model, and thus could achieve an asynchronous acceleration while avoiding memory costs of the backpropagation.

Moreover, in the above task, we actually only need to handle a sequence of stochastic loss functions. Note that Agarwal and Duchi [2011] have shown that the delay only increases the regret of OCO with stochastic and smooth loss functions in an additive way, i.e., an $O(\sqrt{T} + d^2)$ regret bound. Thus, it is interesting to investigate whether the stochastic setting or an intermediate setting [Chen et al., 2024, Wang et al., 2024a] can make delayed BCO easier in a similar way.

Finally, it is also worth extending other BCO algorithms into the delayed setting, e.g., generalizing the algorithm of Saha and Tewari [2011] to improve the regret for smooth functions. However, a more complicated analysis is required because they utilize additional techniques, e.g., the self-concordant barrier [Nemirovski, 2004].

## B    Proof of Lemma 2

Recall the definition of $\mathbf{x}^*$ and $\tilde{\mathbf{x}}^*$ in the proof of Theorem 1. First, it is easy to verify that

$$
\begin{aligned}
\text{Reg}(T) &= \sum_{m=1}^{T/K} \sum_{t=(m-1)K+1}^{mK} \left( f_t(\mathbf{y}_m + \delta\mathbf{u}_t) - f_t(\mathbf{x}^*) \right) \\
&\leq \sum_{m=1}^{T/K} \sum_{t=(m-1)K+1}^{mK} \left( f_t(\mathbf{y}_m) + G\|\delta\mathbf{u}_t\|_2 + f_t(\tilde{\mathbf{x}}^*) - G\left\|\frac{\delta}{r}\mathbf{x}^*\right\|_2 \right) \quad (20) \\
&\leq \sum_{m=1}^{T/K} \sum_{t=(m-1)K+1}^{mK} \left( f_t(\mathbf{y}_m) - f_t(\tilde{\mathbf{x}}^*) \right) + \delta GT + \frac{\delta GRT}{r}
\end{aligned}
$$

where the first inequality is due to Assumption 2, and the second inequality is due to Assumption 1 and $\mathbf{u}_t \sim \mathcal{S}^n$. Note that $\mathbf{y}_1, \ldots, \mathbf{y}_{T/K}$ in Algorithm 1 are computed according to approximate gradients of the $\delta$-smoothed version of original functions, i.e., $\hat{f}_{t,\delta}(\mathbf{x}) = \mathbb{E}_{\mathbf{u}\sim\mathcal{B}^n}[f_t(\mathbf{x} + \delta\mathbf{u})], \forall t \in [T]$. Therefore, before utilizing the unbiasedness of these approximate gradients in Lemma 1, we introduce the following lemma regarding the connection between the original function and its $\delta$-smoothed version over $\mathcal{K}_\delta$ defined in (3).

**Lemma 7.** *(Lemma 2.6 in Hazan [2016]) Let $f(\mathbf{x}) : \mathbb{R}^n \to \mathbb{R}$ be $\alpha$-strongly convex and $G$-Lipschitz over a set $\mathcal{K}$ satisfying Assumption 1. Its $\delta$-smoothed version $\hat{f}_\delta(\mathbf{x})$ defined in (2) has the following properties:*

- *$\hat{f}_\delta(\mathbf{x})$ is $\alpha$-strongly convex over $\mathcal{K}_\delta$;*
- *$|\hat{f}_\delta(\mathbf{x}) - f(\mathbf{x})| \leq \delta G$ for any $\mathbf{x} \in \mathcal{K}_\delta$;*
- *$\hat{f}_\delta(\mathbf{x})$ is $G$-Lipschitz over $\mathcal{K}_\delta$.*

Combining (20) with the second property in Lemma 7, it is easy to verify that

$$
\text{Reg}(T) \leq \sum_{m=1}^{T/K} \sum_{t=(m-1)K+1}^{mK} \left( \hat{f}_{t,\delta}(\mathbf{y}_m) - \hat{f}_{t,\delta}(\tilde{\mathbf{x}}^*) + 2\delta G \right) + \delta GT + \frac{\delta GRT}{r}. \quad (21)
$$

Then, combining (21) with the first property in Lemma 7 where $\alpha = 0$, we have

$$
\text{Reg}(T) \leq \sum_{m=1}^{T/K} \sum_{t=(m-1)K+1}^{mK} \langle \nabla\hat{f}_{t,\delta}(\mathbf{y}_m), \mathbf{y}_m - \tilde{\mathbf{x}}^* \rangle + 3\delta GT + \frac{\delta GRT}{r}.
$$

From the above inequality, it is easy to verify that

$$\text{Reg}(T) \leq \sum_{m=1}^{T/K} \sum_{t=(m-1)K+1}^{mK} \langle \nabla \hat{f}_{t,\delta}(\mathbf{y}_m), \mathbf{y}_m^* - \tilde{\mathbf{x}}^* + \mathbf{y}_m - \mathbf{y}_m^* \rangle + 3\delta GT + \frac{\delta GRT}{r}$$

$$\leq \sum_{m=1}^{T/K} \sum_{t=(m-1)K+1}^{mK} \langle \nabla \hat{f}_{t,\delta}(\mathbf{y}_m), \mathbf{y}_m^* - \tilde{\mathbf{x}}^* \rangle + KG \sum_{m=1}^{T/K} \|\mathbf{y}_m - \mathbf{y}_m^*\|_2 + 3\delta GT + \frac{\delta GRT}{r}$$

(22)

where the last inequality is due to the last property in Lemma 7.

Moreover, according to Lemma 1, we have

$$\sum_{t=(m-1)K+1}^{mK} \mathbb{E}\left[\langle \nabla \hat{f}_{t,\delta}(\mathbf{y}_m), \mathbf{y}_m^* - \tilde{\mathbf{x}}^* \rangle\right] = \sum_{t=(m-1)K+1}^{mK} \mathbb{E}\left[\left\langle \frac{n}{\delta} f_t(\mathbf{y}_m + \delta \mathbf{u}_t)\mathbf{u}_t, \mathbf{y}_m^* - \tilde{\mathbf{x}}^* \right\rangle\right]$$

$$= \mathbb{E}\left[\langle \nabla_m, \mathbf{y}_m^* - \tilde{\mathbf{x}}^* \rangle\right].$$

Finally, we can complete this proof by first taking expectations of both sides in (22) and then substituting the above equality into the right side.

## C  Proof of Lemmas 5 and 6

Lemma 5 can be proved by simply following the proof of Lemma 5 in Garber and Kretzu [2020]. In the following, we first prove Lemma 6, and then include a simple proof of Lemma 5 for completeness.

For brevity, let $\mathbf{g}_t = \frac{n}{\delta} f_t(\mathbf{x}_t)\mathbf{u}_t$ for any $t \in [T]$. Since $\mathbf{g}_1, \dots, \mathbf{g}_{(m-1)K-d+1}$ must be available at the end of round $(m-1)K$, it is not hard to verify that

$$\left\|\sum_{t\in\mathcal{U}_m} \mathbf{g}_t\right\|_2^2 = \left\|\sum_{k=m-1-\lceil d/K \rceil}^{m-1} \sum_{t\in\mathcal{A}_k} \mathbf{g}_t\right\|_2^2 \leq \left(\left\lceil \frac{d}{K} \right\rceil + 1\right) \sum_{k=m-1-\lceil d/K \rceil}^{m-1} \left\|\sum_{t\in\mathcal{A}_k} \mathbf{g}_t\right\|_2^2 \quad (23)$$

where $\mathcal{A}_k = \{(k-1)K+1, \dots, kK\} \cap \mathcal{U}_m$.

Because of $|\mathcal{A}_k| \leq K$, for any $k = m-1-\lceil d/K \rceil, \dots, m-1$, we have

$$\mathbb{E}\left[\left\|\sum_{t\in\mathcal{A}_k} \mathbf{g}_t\right\|_2^2\right] = \mathbb{E}\left[\sum_{t\in\mathcal{A}_k} \|\mathbf{g}_t\|_2^2 + \sum_{i,j\in\mathcal{A}_k, i\neq j} \langle \mathbf{g}_i, \mathbf{g}_j \rangle\right]$$

$$\leq |\mathcal{A}_k| \left(\frac{nM}{\delta}\right)^2 + \mathbb{E}\left[\sum_{i,j\in\mathcal{A}_k, i\neq j} \langle \mathbb{E}[\mathbf{g}_i|\mathbf{y}_k], \mathbb{E}[\mathbf{g}_j|\mathbf{y}_k] \rangle\right]$$

$$\leq K \left(\frac{nM}{\delta}\right)^2 + \mathbb{E}\left[\sum_{i,j\in\mathcal{A}_k, i\neq j} \|\mathbb{E}[\mathbf{g}_i|\mathbf{y}_k]\|_2 \|\mathbb{E}[\mathbf{g}_j|\mathbf{y}_k]\|_2\right]$$

$$\leq K \left(\frac{nM}{\delta}\right)^2 + (|\mathcal{A}_k|^2 - |\mathcal{A}_k|)G^2 \leq K \left(\frac{nM}{\delta}\right)^2 + K^2 G^2$$

(24)

where the first inequality is due to Assumption 3, and the third inequality is due to Assumption 2, Lemma 1, and the last property in Lemma 7.

Combining (23) with (24), we have

$$\mathbb{E}\left[\left\|\sum_{t\in\mathcal{U}_m} \mathbf{g}_t\right\|_2^2\right] \leq 2\left(\frac{d^2}{K^2} + 4\right)\left(K \left(\frac{nM}{\delta}\right)^2 + K^2 G^2\right)$$

which completes the proof of Lemma 6.

Additionally, let $\mathcal{T}_m = \{(m-1)K+1, \ldots, mK\}$. Following (24), it is easy to verify that

$$\mathbb{E}\left[\|\nabla_m\|_2^2\right] = \mathbb{E}\left[\sum_{t \in \mathcal{T}_m} \|\mathbf{g}_t\|_2^2 + \sum_{i,j \in \mathcal{T}_m, i \neq j} \langle \mathbf{g}_i, \mathbf{g}_j \rangle\right] \leq K\left(\frac{nM}{\delta}\right)^2 + K^2 G^2 \tag{25}$$

which completes the proof of Lemma 5.

## D   Proof of Theorem 2

This proof is similar to that of Theorem 1, but requires some specific extensions to utilize the strong convexity. Note that by combining (21) in the proof of Lemma 2 with the strong convexity, we have

$$\mathrm{Reg}(T) \leq \sum_{m=1}^{T/K} \sum_{t \in \mathcal{T}_m} \left(\langle \nabla \hat{f}_{t,\delta}(\mathbf{y}_m), \mathbf{y}_m - \tilde{\mathbf{x}}^* \rangle - \frac{\alpha}{2}\|\mathbf{y}_m - \tilde{\mathbf{x}}^*\|_2^2\right) + 3\delta GT + \frac{\delta GRT}{r} \tag{26}$$

where $\mathcal{T}_m = \{(m-1)K+1, \ldots, mK\}$, $\tilde{\mathbf{x}}^* = (1-\delta/r)\mathbf{x}^*$, and $\mathbf{x}^* \in \mathrm{argmin}_{\mathbf{x} \in \mathcal{K}} \sum_{t=1}^T f_t(\mathbf{x})$.

Moreover, by reorganizing (26) and inserting the ideal action $\mathbf{y}_m^*$,[4] we have

$$\mathrm{Reg}(T) - \left(3\delta GT + \frac{\delta GRT}{r}\right)$$

$$\leq \sum_{m=1}^{T/K} \sum_{t \in \mathcal{T}_m} \left(\langle \nabla \hat{f}_{t,\delta}(\mathbf{y}_m), \mathbf{y}_m^* - \tilde{\mathbf{x}}^* \rangle + \langle \nabla \hat{f}_{t,\delta}(\mathbf{y}_m), \mathbf{y}_m - \mathbf{y}_m^* \rangle - \frac{\alpha}{2}\|\mathbf{y}_m - \tilde{\mathbf{x}}^*\|_2^2\right) \tag{27}$$

$$\leq \sum_{m=1}^{T/K} \sum_{t \in \mathcal{T}_m} \left(\langle \nabla \hat{f}_{t,\delta}(\mathbf{y}_m), \mathbf{y}_m^* - \tilde{\mathbf{x}}^* \rangle - \frac{\alpha}{2}\|\mathbf{y}_m - \tilde{\mathbf{x}}^*\|_2^2\right) + \sum_{m=1}^{T/K} KG\|\mathbf{y}_m - \mathbf{y}_m^*\|_2$$

where the last equality is due to Assumption 2 and the last property in Lemma 7.

For brevity, we define surrogate losses as $\ell_m(\mathbf{x}) = \langle \nabla_m, \mathbf{x} \rangle + \frac{\alpha K}{2}\|\mathbf{y}_m - \mathbf{x}\|_2^2$ for $m \in [T/K]$, and notice that

$$\mathbf{y}_{m+1}^* = \mathrm{argmin}_{\mathbf{x} \in \mathcal{K}_\delta} \sum_{i=1}^m \ell_i(\mathbf{x}) \tag{28}$$

in the strongly convex case. Then, it is not hard to verify that

$$\mathbb{E}\left[\sum_{m=1}^{T/K} \sum_{t \in \mathcal{T}_m} \left(\langle \nabla \hat{f}_{t,\delta}(\mathbf{y}_m), \mathbf{y}_m^* - \tilde{\mathbf{x}}^* \rangle - \frac{\alpha}{2}\|\mathbf{y}_m - \tilde{\mathbf{x}}^*\|_2^2\right)\right]$$

$$= \mathbb{E}\left[\sum_{m=1}^{T/K} \sum_{t \in \mathcal{T}_m} \left(\langle \frac{n}{\delta} f_t(\mathbf{y}_m + \delta \mathbf{u}_t)\mathbf{u}_t, \mathbf{y}_m^* - \tilde{\mathbf{x}}^* \rangle - \frac{\alpha}{2}\|\mathbf{y}_m - \tilde{\mathbf{x}}^*\|_2^2\right)\right]$$

$$= \mathbb{E}\left[\sum_{m=1}^{T/K} \left(\langle \nabla_m, \mathbf{y}_{m+1}^* - \tilde{\mathbf{x}}^* + \mathbf{y}_m^* - \mathbf{y}_{m+1}^* \rangle - \frac{\alpha K}{2}\|\mathbf{y}_m - \tilde{\mathbf{x}}^*\|_2^2\right)\right] \tag{29}$$

$$\leq \mathbb{E}\left[\sum_{m=1}^{T/K} \left(\ell_m(\mathbf{y}_{m+1}^*) - \ell_m(\tilde{\mathbf{x}}^*)\right)\right] + \mathbb{E}\left[\sum_{m=1}^{T/K} \|\nabla_m\|_2\|\mathbf{y}_m^* - \mathbf{y}_{m+1}^*\|_2\right]$$

$$\leq \mathbb{E}\left[\sum_{m=1}^{T/K} \|\nabla_m\|_2\|\mathbf{y}_m^* - \mathbf{y}_{m+1}^*\|_2\right]$$

where the first equality is due to Lemma 1, and the last inequality is due to (28) and Lemma 3. From (27) and (29), we still need to bound $\|\mathbf{y}_m - \mathbf{y}_m^*\|_2$ and $\|\mathbf{y}_m^* - \mathbf{y}_{m+1}^*\|_2$.

---

[4]Note that the definition of $\mathbf{y}_m^*$ in (5) for strongly convex functions is only valid for $m \geq 2$. For $m = 1$, we simply set $\mathbf{y}_1^* = \mathbf{y}_1$.

To this end, we notice that $\mathbf{y}_m^*$ for any $m = 2, \ldots, T/K$ is equal to

$$\mathbf{y}_m^* = \underset{\mathbf{x} \in \mathcal{K}_\delta}{\operatorname{argmin}} \left\{ \left\langle \sum_{i=1}^{m-1} (\nabla_i - \alpha K \mathbf{y}_i), \mathbf{x} \right\rangle + \frac{\alpha(m-1)K}{2} \|\mathbf{x}\|_2^2 \right\}. \tag{30}$$

Similarly, for any $m = 2, \ldots, T/K$, the action $\mathbf{y}_m$ of Algorithm 1 with $\alpha > 0$ is equal to

$$\mathbf{y}_m = \underset{\mathbf{x} \in \mathcal{K}_\delta}{\operatorname{argmin}} \left\{ \left\langle \bar{\mathbf{g}}_{(m-1)K} - \sum_{i=1}^{m-1} \alpha K \mathbf{y}_i, \mathbf{x} \right\rangle + \frac{\alpha(m-1)K}{2} \|\mathbf{x}\|_2^2 \right\}. \tag{31}$$

Combining (30) and (31) with Lemma 4, for any $m = 2, \ldots, T/K$, we have

$$\|\mathbf{y}_m - \mathbf{y}_m^*\|_2 \leq \frac{1}{\alpha(m-1)K} \left\| \bar{\mathbf{g}}_{(m-1)K} - \sum_{i=1}^{m-1} \nabla_i \right\|_2 = \frac{1}{\alpha(m-1)K} \left\| \sum_{t \in \mathcal{U}_m} \frac{n}{\delta} f_t(\mathbf{x}_t) \mathbf{u}_t \right\|_2 \tag{32}$$

where $\mathcal{U}_m$ is defined in (18).

Moreover, from (1), for any $m = 2, \ldots, T/K$, we have

$$\|\mathbf{y}_m^* - \mathbf{y}_{m+1}^*\|_2^2 \leq \frac{2}{\alpha m K} \left( \sum_{i=1}^m \ell_i(\mathbf{y}_m^*) - \sum_{i=1}^m \ell_i(\mathbf{y}_{m+1}^*) \right)$$

$$\leq \frac{2}{\alpha m K} \left( \ell_m(\mathbf{y}_m^*) - \ell_m(\mathbf{y}_{m+1}^*) \right)$$

$$\leq \frac{2}{\alpha m K} \left( \langle \nabla_m + \alpha K(\mathbf{y}_m^* - \mathbf{y}_m), \mathbf{y}_m^* - \mathbf{y}_{m+1}^* \rangle \right)$$

$$\leq \frac{2}{\alpha(m-1)K} \left( \|\nabla_m\|_2 + 2\alpha K R \right) \|\mathbf{y}_m^* - \mathbf{y}_{m+1}^*\|_2$$

where the first inequality is due to the definition of $\mathbf{y}_m^*$, and the last inequality is due to Assumption 1. The above inequality further implies that

$$\|\mathbf{y}_m^* - \mathbf{y}_{m+1}^*\|_2 \leq \frac{2}{\alpha(m-1)K} \left( \|\nabla_m\|_2 + 2\alpha K R \right). \tag{33}$$

Combining (27), (29), (32), and (33), we have

$$\mathbb{E}\left[\operatorname{Reg}(T)\right] \leq \mathbb{E}\left[ \sum_{m=1}^{T/K} \|\nabla_m\|_2 \|\mathbf{y}_m^* - \mathbf{y}_{m+1}^*\|_2 \right] + \sum_{m=1}^{T/K} K G \mathbb{E}\left[\|\mathbf{y}_m - \mathbf{y}_m^*\|_2\right] + 3\delta G T + \frac{\delta G R T}{r}$$

$$\leq \mathbb{E}[\|\nabla_1\|_2 \|\mathbf{y}_1 - \mathbf{y}_2^*\|_2] + \mathbb{E}\left[ \sum_{m=2}^{T/K} \frac{2\left(\|\nabla_m\|_2^2 + 2\alpha K R \|\nabla_m\|_2\right)}{\alpha(m-1)K} \right]$$

$$+ \frac{G}{\alpha(m-1)} \sum_{m=2}^{T/K} \mathbb{E}\left[ \left\| \sum_{t \in \mathcal{U}_m} \frac{n}{\delta} f_t(\mathbf{x}_t) \mathbf{u}_t \right\|_2 \right] + 3\delta G T + \frac{\delta G R T}{r}.$$

Combining the above inequality with Assumption 1, Lemma 5, Lemma 6, and $\gamma = K \left( \frac{nM}{\delta} \right)^2 + K^2 G^2$, it is easy to verify that

$$\mathbb{E}\left[\operatorname{Reg}(T)\right] \leq 2R\sqrt{\gamma} + \sum_{m=2}^{T/K} \frac{1}{m-1} \left( \frac{2\gamma}{\alpha K} + 4R\sqrt{\gamma} + \frac{G}{\alpha} \sqrt{2 \left( \frac{d^2}{K^2} + 4 \right) \gamma} \right) + 3\delta G T + \frac{\delta G R T}{r}$$

$$\leq 2R\sqrt{\gamma} + (1 + \ln T) \left( \frac{2\gamma}{\alpha K} + 4R\sqrt{\gamma} + \frac{G}{\alpha} \sqrt{2 \left( \frac{d^2}{K^2} + 4 \right) \gamma} \right) + 3\delta G T + \frac{\delta G R T}{r}$$

$$= \frac{2\gamma C_T}{\alpha K} + C_T' R\sqrt{\gamma} + \frac{G C_T}{\alpha} \sqrt{2 \left( \frac{d^2}{K^2} + 4 \right) \gamma} + 3\delta G T + \frac{\delta G R T}{r}$$

where $C_T = 1 + \ln T$ and $C_T' = 6 + 4\ln T$.

# E    Proof of Theorem 3

The main idea of this proof is to combine the proof of Theorem 2 with an improved property of the $\delta$-smoothed version of smooth functions [Agarwal et al., 2010].

Specifically, for any $t \in [T]$ and $\mathbf{x}$, according to the smoothness of functions, we have

$$\hat{f}_{t,\delta}(\mathbf{x}) \leq \mathbb{E}_{\mathbf{u} \sim \mathcal{B}^n}\left[f_t(\mathbf{x}) + \langle \nabla f_t(\mathbf{x}), \delta\mathbf{u}\rangle + \frac{\beta\delta^2\|\mathbf{u}\|_2^2}{2}\right] = f_t(\mathbf{x}) + \frac{\beta\delta^2}{2} \tag{34}$$

where $\hat{f}_{t,\delta}(\mathbf{x}) = \mathbb{E}_{\mathbf{u} \sim \mathcal{B}^n}[f_t(\mathbf{x} + \delta\mathbf{u})]$ and the last equality is due to $\mathbb{E}_{\mathbf{u} \sim \mathcal{B}^n}[\mathbf{u}] = \mathbf{0}$.

Moreover, due to the convexity of functions, for any $t \in [T]$ and $\mathbf{x}$, we have

$$\hat{f}_{t,\delta}(\mathbf{x}) \geq \mathbb{E}_{\mathbf{u} \sim \mathcal{B}^n}[f_t(\mathbf{x}) + \langle \nabla f_t(\mathbf{x}), \delta\mathbf{u}\rangle] = f_t(\mathbf{x}). \tag{35}$$

Then, let $\mathbf{x}^* = \operatorname{argmin}_{\mathbf{x} \in \mathbb{R}^n} \sum_{t=1}^{T} f_t(\mathbf{x})$ and $\tilde{\mathbf{x}}^* = (1 - \delta/r)\mathbf{x}^*$, where $r = 2G/\alpha$. According to (10), we have $\mathbf{x}^* \in \mathcal{K}'$ and $\tilde{\mathbf{x}}^* \in \mathcal{K}'_\delta$, where $\mathcal{K}'$ and $\mathcal{K}'_\delta$ are defined in (11) and (12), respectively. By further defining $\mathcal{T}_m = \{(m-1)K + 1, \ldots, mK\}$, it is not hard to verify that

$$
\begin{aligned}
\mathbb{E}[\operatorname{Reg}(T)] =& \mathbb{E}\left[\sum_{m=1}^{T/K}\sum_{t\in\mathcal{T}_m}(f_t(\mathbf{y}_m + \delta\mathbf{u}_t) - f_t(\mathbf{x}^*))\right] \\
\leq& \mathbb{E}\left[\sum_{m=1}^{T/K}\sum_{t\in\mathcal{T}_m}\left(f_t(\mathbf{y}_m) + \langle\nabla f_t(\mathbf{y}_m), \delta\mathbf{u}_t\rangle + \frac{\beta\delta^2\|\mathbf{u}_t\|_2^2}{2}\right)\right] \\
& + \mathbb{E}\left[\sum_{m=1}^{T/K}\sum_{t\in\mathcal{T}_m}\left(-f_t(\tilde{\mathbf{x}}^*) + \left\langle\nabla f_t(\mathbf{x}^*), -\frac{\delta\mathbf{x}^*}{r}\right\rangle + \frac{\beta\delta^2\|\mathbf{x}^*\|_2^2}{2r}\right)\right] \\
=& \mathbb{E}\left[\sum_{m=1}^{T/K}\sum_{t\in\mathcal{T}_m}\left(f_t(\mathbf{y}_m) - f_t(\tilde{\mathbf{x}}^*) + \frac{\beta\delta^2}{2} + \frac{\beta\delta^2 G}{\alpha}\right)\right] \\
\leq& \mathbb{E}\left[\sum_{m=1}^{T/K}\sum_{t\in\mathcal{T}_m}\left(\hat{f}_{t,\delta}(\mathbf{y}_m) - \hat{f}_{t,\delta}(\tilde{\mathbf{x}}^*)\right)\right] + \beta\delta^2 T + \frac{\beta\delta^2 GT}{\alpha}
\end{aligned}
\tag{36}
$$

where the first inequality is due to the smoothness of functions, and the last inequality is due to (34) and (35).

Then, we follow the definition of $\mathbf{y}_m^*$ in (28), but replace $\mathcal{K}_\delta$ utilized in (28) with $\mathcal{K}'_\delta$. Combining (36) with the strong convexity of functions, we have

$$
\begin{aligned}
&\mathbb{E}[\operatorname{Reg}(T)] - \left(\beta\delta^2 T + \frac{\beta\delta^2 GT}{\alpha}\right) \\
\leq& \mathbb{E}\left[\sum_{m=1}^{T/K}\sum_{t\in\mathcal{T}_m}\left(\langle\nabla\hat{f}_{t,\delta}(\mathbf{y}_m), \mathbf{y}_m - \tilde{\mathbf{x}}^*\rangle - \frac{\alpha}{2}\|\mathbf{y}_m - \tilde{\mathbf{x}}^*\|_2^2\right)\right] \\
=& \mathbb{E}\left[\sum_{m=1}^{T/K}\sum_{t\in\mathcal{T}_m}\left(\langle\nabla\hat{f}_{t,\delta}(\mathbf{y}_m), \mathbf{y}_m^* - \tilde{\mathbf{x}}^*\rangle + \langle\nabla\hat{f}_{t,\delta}(\mathbf{y}_m), \mathbf{y}_m - \mathbf{y}_m^*\rangle - \frac{\alpha}{2}\|\mathbf{y}_m - \tilde{\mathbf{x}}^*\|_2^2\right)\right] \\
\leq& \mathbb{E}\left[\sum_{m=1}^{T/K}\sum_{t\in\mathcal{T}_m}\left(\langle\nabla\hat{f}_{t,\delta}(\mathbf{y}_m), \mathbf{y}_m^* - \tilde{\mathbf{x}}^*\rangle - \frac{\alpha}{2}\|\mathbf{y}_m - \tilde{\mathbf{x}}^*\|_2^2\right)\right] + \mathbb{E}\left[\sum_{m=1}^{T/K}KG\|\mathbf{y}_m - \mathbf{y}_m^*\|_2\right]
\end{aligned}
\tag{37}
$$

where we simply set $\mathbf{y}_1^* = \mathbf{y}_1$, and the last inequality is due to Assumption 2 and the last property in Lemma 7.

Let $R = 2G/\alpha$ denote the radius of $\mathcal{K}'$. It is not hard to verify that (29), (32), and (33) in the proof of Theorem 2 still hold here. Therefore, we have

$$
\begin{aligned}
&\mathbb{E}\left[\text{Reg}(T)\right] - \left(\beta\delta^2 T + \frac{\beta\delta^2 GT}{\alpha}\right)\\
&\leq \mathbb{E}\left[\sum_{m=1}^{T/K} \|\nabla_m\|_2 \|\mathbf{y}_m^* - \mathbf{y}_{m+1}^*\|_2\right] + \mathbb{E}\left[\sum_{m=1}^{T/K} KG\|\mathbf{y}_m - \mathbf{y}_m^*\|_2\right]\\
&\leq \mathbb{E}[\|\nabla_1\|_2\|\mathbf{y}_1 - \mathbf{y}_2^*\|_2] + \mathbb{E}\left[\sum_{m=2}^{T/K} \frac{2\left(\|\nabla_m\|_2^2 + 2\alpha KR\|\nabla_m\|_2\right)}{\alpha(m-1)K}\right]\\
&\quad + \frac{G}{\alpha(m-1)} \sum_{m=2}^{T/K} \mathbb{E}\left[\left\|\sum_{t\in\mathcal{U}_m} \frac{n}{\delta} f_t(\mathbf{x}_t)\mathbf{u}_t\right\|_2\right]
\end{aligned}
\tag{38}
$$

where the first inequality is due to (37) and (29), and the last one is due to (32) and (33).

Finally, combining (38) with Lemma 5, Lemma 6, and $\gamma = K\left(\frac{nM}{\delta}\right)^2 + K^2 G^2$, we have

$$
\begin{aligned}
\mathbb{E}\left[\text{Reg}(T)\right] &\leq 2R\sqrt{\gamma} + \sum_{m=2}^{T/K} \frac{1}{m-1}\left(\frac{2\gamma}{\alpha K} + 4R\sqrt{\gamma} + \frac{G}{\alpha}\sqrt{2\left(\frac{d^2}{K^2}+4\right)\gamma}\right) + \beta\delta^2 T + \frac{\beta\delta^2 GT}{\alpha}\\
&\leq 2R\sqrt{\gamma} + (1+\ln T)\left(\frac{2\gamma}{\alpha K} + 4R\sqrt{\gamma} + \frac{G}{\alpha}\sqrt{2\left(\frac{d^2}{K^2}+4\right)\gamma}\right) + \beta\delta^2 T + \frac{\beta\delta^2 GT}{\alpha}\\
&= \frac{2\gamma C_T}{\alpha K} + \frac{2C_T' G\sqrt{\gamma}}{\alpha} + \frac{GC_T}{\alpha}\sqrt{2\left(\frac{d^2}{K^2}+4\right)\gamma} + \beta\delta^2 T + \frac{\beta\delta^2 GT}{\alpha}
\end{aligned}
$$

where the last equality is due to $R = 2G/\alpha$, $C_T = 1 + \ln T$, and $C_T' = 6 + 4\ln T$.

## F A refined regret bound for Bistritz et al. [2022]

From Theorem 4 of Bistritz et al. [2022], their algorithm can achieve the following regret bound

$$
\mathbb{E}[\text{Reg}(T)] = O\left(\delta T + \frac{\eta n^2 T}{\delta^2} + \frac{1}{\eta} + \frac{n\eta\bar{d}T}{\delta}\right)
\tag{39}
$$

for BCO with delayed feedback, where $\delta > 0$ and $\eta > 0$ denote the exploration radius and the step size, respectively. Then, by further substituting

$$
\delta = \max\left\{T^{-1/4}, T^{-1/3}\bar{d}^{1/3}\right\} \text{ and } \eta = \min\left\{n^{-1}T^{-3/4}, n^{-1/2}T^{-2/3}\bar{d}^{-1/3}\right\}
$$

into (39), Bistritz et al. [2022] have established the $O(nT^{3/4} + \sqrt{n}\bar{d}^{1/3}T^{2/3})$ regret bound. However, we notice that

$$
\min_{\delta>0,\eta>0} O\left(\delta T + \frac{\eta n^2 T}{\delta^2} + \frac{1}{\eta} + \frac{n\eta\bar{d}T}{\delta}\right) = \min_{\delta>0} O\left(\delta T + \sqrt{\frac{n^2 T}{\delta^2} + \frac{n\bar{d}T}{\delta}}\right)
\tag{40}
$$

where the equality holds with

$$
\eta = \left(\frac{n^2 T}{\delta^2} + \frac{n\bar{d}T}{\delta}\right)^{-1/2}.
\tag{41}
$$

From (40), if $n^2 T\delta^{-2} \geq n\bar{d}T\delta^{-1}$, we have

$$
\min_{\delta>0,\eta>0} O\left(\delta T + \frac{\eta n^2 T}{\delta^2} + \frac{1}{\eta} + \frac{n\eta\bar{d}T}{\delta}\right) = \min_{\delta>0} O\left(\delta T + \frac{n\sqrt{T}}{\delta}\right) = O\left(\sqrt{n}T^{3/4}\right)
\tag{42}
$$

where the last equality holds with $\delta = \sqrt{n}T^{-1/4}$.

Otherwise, combining (40) with $n^2 T \delta^{-2} < n\bar{d} T \delta^{-1}$, we have

$$\min_{\delta > 0, \eta > 0} O\left(\delta T + \frac{\eta n^2 T}{\delta^2} + \frac{1}{\eta} + \frac{n\eta\bar{d}T}{\delta}\right) = \min_{\delta > 0} O\left(\delta T + \sqrt{\frac{n\bar{d}T}{\delta}}\right) = O\left((n\bar{d})^{1/3} T^{2/3}\right) \quad (43)$$

where the last equality holds with $\delta = (n\bar{d})^{1/3} T^{-1/3}$.

Combining (39) with (41), (42), and (43), we can improve the regret bound of Bistritz et al. [2022] to

$$\mathbb{E}[\text{Reg}(T)] = O\left(\sqrt{n}T^{3/4} + (n\bar{d})^{1/3} T^{2/3}\right) \quad (44)$$

by setting $\delta$ and $\eta$ as

$$\delta = \max\left\{\sqrt{n}T^{-1/4}, (n\bar{d})^{1/3} T^{-1/3}\right\} \text{ and } \eta = \min\left\{n^{-1/2} T^{-3/4}, (n\bar{d})^{-1/3} T^{-2/3}\right\}.$$

