# OpenReview forum: "Improved Regret for Bandit Convex Optimization with Delayed Feedback"
_NeurIPS.cc/2024/Conference — NeurIPS 2024 poster_

### Official Review · Reviewer_GZvs · 2024-07-06

**Soundness:** 3
**Presentation:** 2
**Contribution:** 3
**Rating:** 6
**Confidence:** 4

**Summary:**

The paper investigates the online convex optimization problem with delayed bandit feedback.

The main contribution is introducing a new algorithm that uses a block updating mechanism with FTRL, proving the algorithm achieves a delay-dependent regret of $O(\sqrt{dT})$, which is known to be optimal when the average delay is close to the maximal delay: $\bar{d} \approx d$.

**Strengths:**

* The paper shows a valuable algorithm that improves on the known regret bounds for the delayed BCO.
* The techniques used, specifically block updates, while not novel, are original for this kind of problem.

**Weaknesses:**

* The paper could be written more clearly. I found myself understanding parts of the introduction only after going through the entire paper (e.g. line 69).
* The proofs are hard to follow. Some explanations before each lemma to understand how those are used would be helpful.
* It seems hard to use the blocking updates to further improve the regret, so the value of the presented algorithm could be temporary.

**Questions:**

Can you explain each term in the theorems and where does it come from? As is, the theorems are packed with different terms and besides substituting the optimal values I don't know how to interpret them

**Limitations:**

The authors properly acknowledge the limitations of their work.

---

> ### Author Rebuttal · Authors · 2024-08-04
>
> Many thanks for the constructive reviews!
>
> ---
> Q1: The paper could be written more clearly. I found myself understanding parts of the introduction only after going through the entire paper (e.g. line 69).
>
> A1: Thank you for the helpful suggestion. We will improve our writing by adding necessary explanations to help the understanding.
>
> ---
> Q2: The proofs are hard to follow. Some explanations before each lemma to understand how those are used would be helpful.
>
> A2: As you suggested, we will explain the usefulness of each lemma in our analysis. Moreover, we will start each proof by providing a high-level overview of the upcoming procedures.
>
> ---
> Q3: It seems hard to use the blocking updates to further improve the regret, so the value of the presented algorithm could be temporary.
>
> A3: We agree that the blocking update mechanism may prevent our algorithm from further improving regret bounds to be depending on the average delay. However, we want to emphasize that our regret bounds achieved via the blocking update mechanism are already quite favorable,  and it is highly non-trivial to develop a better algorithm, especially in the case with strongly convex functions. Note that even in the easier full-information setting, previous studies can only achieve the $O(d\log T)$ regret for strongly convex functions, which is the same as the delay-dependent part in our regret bounds for strongly convex functions.
>
> ---
> Q4: Can you explain each term in the theorems and where does it come from? As is, the theorems are packed with different terms and besides substituting the optimal values I don't know how to interpret them
>
> A4: We first provide the following explanations for each term in our Theorem 1.
> 1) The first two terms $\frac{4R^2}{\eta}+\frac{\eta T\gamma}{2K}$ are an upper bound on the expected regret of an ideal action for each block, i.e., $\mathbf{y}_m^\ast$ defined in line 305 of our paper. Therefore, they are affected by the step size $\eta$, the block size $K$, and the variance of cumulative gradients in each block $\gamma$. Moreover, the variance $\gamma$ depends on both the block size $K$ and the exploration radius $\delta$ at first glance, but we can select an appropriate $K$ to remove the extra dependence on $\delta$ as discussed in our paper.
> 2) The third term $\frac{\eta TG}{2}\sqrt{2\left(\frac{d^2}{K^2}+4\right)\gamma}$ is an upper bound on the expected cumulative distance between the preparatory action $\mathbf{y}_m$ of our Algorithm 1 and the ideal one $\mathbf{y}_m^\ast$, which is further affected by the maximum delay $d$.
> 3) The last two terms $3\delta GT+\frac{\delta GRT}{r}$ are caused by the error of the shrunk set $\\mathcal{K}\_\\delta$ and the $\delta$-smoothed function $\hat{f}_{t,\delta}(\cdot)$, and thus are affected by the exploration radius $\delta$.
>
> Additionally, we note that the terms in our Theorems 2 and 3 can be similarly divided into these three categories. In the revised version, we will provide detailed explanations for each term in all our theorems.

---

> > ### Comment · Reviewer_GZvs · 2024-08-09
> >
> > Thank you, I am keeping my score.

---

### Official Review · Reviewer_knrh · 2024-07-12

**Soundness:** 3
**Presentation:** 4
**Contribution:** 3
**Rating:** 7
**Confidence:** 3

**Summary:**

The paper investigates the problem of bandit convex optimization (BCO) with delayed feedback, where the value of the action is revealed after some delay. The authors proposed an algorithm D-FTBL, and proved that it enjoys a regret bound of $O\left(\sqrt{n} T^{3/4}+\sqrt{d T}\right)$, closing the gap between the previous result and the lower bound on delay-dependent part. Furthermore, the proposed algorithm can improve the regret bound to $O\left((n T)^{2 / 3} \log ^{1 / 3} T+d \log T\right)$ for strongly convex functions, and if the action sets are unconstrained, the proposed algorithm can achieve an $O(n \sqrt{T \log T}+d \log T)$ regret bound for strongly convex and smooth functions.

**Strengths:**

- The writing style of the article is excellent, and the overall flow is very smooth and enjoyable to follow.
- The literature review is thorough and well-integrated into the paper. It provides a solid foundation for the research by situating it within the existing literature of BCO and highlighting the gaps that the authors aim to fill.
- The methodology seems innovative. The application of blocking update mechanism not only adds substantial value to the current study but also potentially inspires other problems with delayed feedback.
-  The theoretical results presented in the paper are logically sound and well-supported. Each point is substantiated with adequate evidence, avoiding any logical leaps or inconsistencies.

**Weaknesses:**

The paper lacks numerical experiments. Besides this, I did not spot any significant weaknesses.

**Questions:**

I would be curious about how much improvement can be made in practice. In bandit setting, people often consider large time horizon, in which cases the regret is dominated by the delay-independent term.

**Limitations:**

The authors addressed the limitations and consider it as future work.

---

> ### Author Rebuttal · Authors · 2024-08-04
>
> Many thanks for the constructive reviews!
>
> ---
> Q: The paper lacks numerical experiments ... I would be curious about how much improvement can be made in practice. In bandit setting, people often consider large time horizon, in which cases the regret is dominated by the delay-independent term.
>
> A: Please check our **common response to the suggestion about experiments**, which introduces numerical results completed during the rebuttal period, and has shown the advantage of our algorithm in practice. Moreover, we want to emphasize that although $T$ could be very large, the delay-dependent term in the existing $O(\sqrt{n}T^{3/4}+(n\bar{d})^{1/3}T^{2/3})$ regret bound [Bistritz et al., 2022] cannot be ignored. Specifically, this regret bound is dominated by the delay-independent term only for $\bar{d}=O(n^{1/2}T^{1/4})$. However, this condition can be easily violated due to the sublinear dependence on $T$ and $n$, and the fact that the delay may also increase for larger $T$. For example, our experiment on ijcnn1 has $T=40000$ and $n=22$, and thus the condition is violated as long as $\bar{d}>n^{1/2}T^{1/4}\approx 67$. Even if we consider a much larger $T=400000000$, $\bar{d}>664$ is sufficient to violate the condition. By contrast, our $O(\sqrt{n}T^{3/4}+\sqrt{dT})$ regret bound is dominated by the delay-independent term for a larger amount of delays, i.e., $d=O(n\sqrt{T})$.

---

> > ### Comment · Reviewer_knrh · 2024-08-09
> >
> > Thank you for the response. I will keep my score positive.

---

### Official Review · Reviewer_pbhB · 2024-07-12

**Soundness:** 3
**Presentation:** 3
**Contribution:** 3
**Rating:** 7
**Confidence:** 3

**Summary:**

This paper studies the problem of bandit convex optimization with delayed feedback (where the feedback for round $t$ is delayed by an arbitrary number of rounds $d_t$).
For this problem, they show $O(\sqrt{n} T^{3/4} + \sqrt{d T})$ regret in general, $O((n T)^{2/3} \log^{1/3} (T) + d \log(T))$ regret for strongly-convex functions, and $O(n \sqrt{T \log(T)} + d \log(T))$ regret for smooth and strongly-convex functions in an unconstrained settting.
In these bounds, $n$ is the dimension, $T$ is the horizon and $d$ is the maximum delay.

**Strengths:**

- Their results for the general setting strictly improve on results that use the maximum delay $d$ (Heliou et al (2020)), and supplement results that use the average delay $\bar{d}$ (Bistritz et al 2022).
- They give improved regret for specific settings.
- The literature review is very thorough and the authors are diligent in pointing to the origin of ideas throughout the paper.
- There is extensive discussion on the difference in techniques with respect to existing approaches and how their approach result in tighter regret gaurantees.
- The presentation is clear and intuitive.

**Weaknesses:**

- The paper only makes an improvement on state-of-the-art for certain delay sequences, i.e. when $d = O((n\bar{d})^{2/3} T^{1/3})$.

**Questions:**

No questions at this point.

**Limitations:**

There is some discussion of the limitation due to the fact that the regret is given in terms of maximum delay rather than average delay.

---

> ### Author Rebuttal · Authors · 2024-08-04
>
> Many thanks for the constructive reviews!
>
> ---
> Q: The paper only makes an improvement on state-of-the-art for certain delay sequences, i.e. when $d=O((n\bar{d})^{2/3}T^{1/3})$.
>
> A: First, we want to emphasize that both $n$ and $T$ can be very large in modern online applications, and thus the condition $d=O((n\bar{d})^{2/3}T^{1/3})$ can be satisfied by many delay sequences including those with $\bar{d}=1$. Second, it is also worth noting that besides convex functions considered in previous studies [Héliou et al., 2020; Bistritz et al., 2022], our paper further investigates two special cases with strongly convex functions, and achieves $O((nT)^{2/3}\log^{1/3}T+d\log T)$ and $O(n\sqrt{T\log T}+d\log T)$ regret bounds, respectively. These two bounds are better than the state-of-the-art result for a larger portion of delay sequences.

---

> > ### Comment · Reviewer_pbhB · 2024-08-07
> > **Thank you for the response**
> >
> > Thanks for the clear response. I'm sticking with my positive score.

---

### Official Review · Reviewer_S7rX · 2024-07-13

**Soundness:** 3
**Presentation:** 3
**Contribution:** 2
**Rating:** 5
**Confidence:** 3

**Summary:**

This paper studies the bandit convex optimization problem with delayed feedback, where the loss value of the selected action is revealed under an arbitrary delay.

Previous work achieves $\mathcal{O}( \sqrt{n} T^{3/4} + (n\bar{d})^{1/3} T^{2/3})$ regret bound of this problem. The authors develop a novel algorithm and show that it improves the delay-related part to $\mathcal{O}( sqrt{dT})$ when $d$ the maximum delay, is close to $\bar{d}$, the average delay (specifically, strictly better when d = $\mathcal{O}( (n\bar{d})^{2/3} T^{1/3}  )$. The authors claim that the primary idea is to decouple the joint effect of the delays by incorporating the delayed bandit feedback with a blocking update mechanism, which reduces the correlation between recent delayed updates (otherwise there could be a $d^2$ term).

**Strengths:**

1. Though I have skimmed the proof of several lemmas, the analysis part seems to be rigorous and mathematically correct.
2. The usage of blocking update mechanism is very interesting, and could be applied in other similar settings.

**Weaknesses:**

1. The contribution of this work is quite concerning. It would be better for the authors to emphasis the contribution (either algorithmic or analytic) on improving the delayed feedback result for BCO problem in certain conditions that the max delay $d$ is close to $\bar{d}$? Some parts of the proof are rather straightforward and standard.

**Questions:**

The questions are raised in the weakness section. I am willing to re-evaluate the scores if these questions are properly answered.

**Limitations:**

This paper is pure theoretical and does not have any limitations.

---

> ### Author Rebuttal · Authors · 2024-08-04
>
> Many thanks for the constructive reviews!
>
> ---
> Q: The contribution of this work is quite concerning. It would be better for the authors to emphasis the contribution (either algorithmic or analytic) on improving the delayed feedback result for BCO problem in certain conditions that the max delay $d$ is close to $\bar{d}$? Some parts of the proof are rather straightforward and standard.
>
> A: Thank you for the helpful suggestion. Our technical contributions can be summarized as follows.
> 1) At the algorithmic level, our paper is the first work that exploits the blocking update mechanism to design improved algorithms for delayed BCO. Moreover, unlike existing algorithms [Héliou et al., 2020; Bistritz et al., 2022] based on online gradient descent, our algorithm is based on follow-the-regularized-leader (FTRL). As discussed in lines 228 to 235 of our paper, in this way, we can utilize the delayed information more elegantly.
> 2) At the analytic level, we derive the first regret bound that can decouple the joint effect of the delays and the bandit feedback. To this end, besides some standard analysis for FTRL and BCO, we need to carefully analyze the delay effect under blocking update to establish an improved upper bound for $\\|\mathbf{y}_m-\mathbf{y}_m^\ast\\|_2$, where $\mathbf{y}_m $ is the preparatory action of our Algorithm 1 and $\mathbf{y}_m^\ast$ is an ideal action defined in line 305 of our paper.
> 3) Moreover, different from previous studies [Héliou et al., 2020; Bistritz et al., 2022] that only consider convex functions, our paper further investigates two special cases with strongly convex functions, and achieves better regret bounds.

---

> > ### Comment · Reviewer_S7rX · 2024-08-13
> >
> > Thank the authors for their response. I would like to keep the current score.

---

### Official Review · Reviewer_vCin · 2024-07-15

**Soundness:** 3
**Presentation:** 4
**Contribution:** 3
**Rating:** 7
**Confidence:** 4

**Summary:**

The authors consider the problem of bandit convex optimization in the adversarial setting under delays. In each round, an adversary selects a convex function $f_t$, the optimizer selects an input $x_t$ and observes $f_t(x_t)$ with a delay of $d_t$ timesteps. The goal is to minimize regret with respect to the best action in hindsight.

Without any delay, the best regret is obtained by the zeroth order one-point flaxman updates. Prior work in the delayed setting adapts this algorithm to use the oldest available feedback to make the update. However, the observation that the authors make is that according to this scheme, information might become available much before it is used -- if there is older information that has not been used yet.

So instead, to minimize the gap between availability and utilization of information, the authors propose a blocking algorithm. Where at the end of each block, all available information so far is used to make the update. This enables them to get improved regret bounds.

**Strengths:**

1. The paper is very well-written with the results laid out very clearly, and the rationale for the solution explained well.
2. The solution to use blocking in order to minimize stale information is creative, and leads to better bounds.
3. The results are comprehensive across different classes of objectives, and the authors thus show the wide applicability of the technique.

**Weaknesses:**

**Significance** The practical impact of this theoretical toy seems limited, and the problem was of more interest a couple of years ago. In order to increase the reception, it would be instructive to connect the implications of these findings to other topics of recent interest or include some simulations

**Questions:**

1. Could the authors comment on what the problem looks like for the regular stochastic case with all $f_t = f$ for some fixed $f$, and how this might make the problem easier under delays?
2. Why do the authors conjecture that the lower bound depends on $\bar{d}$, whereas the upper bound depends on $d$? What improvements would be required to get these to match?

**Limitations:**

Yes

---

> ### Author Rebuttal · Authors · 2024-08-04
>
> Many thanks for the constructive reviews!
>
> ---
> Q1: The practical impact of this theoretical toy seems limited, and the problem was of more interest a couple of years ago. In order to increase the reception, it would be instructive to connect the implications of these findings to other topics of recent interest or include some simulations
>
> A1: Thanks for the suggestion. Please check our **common response to the suggestion about experiments**, which introduces numerical results completed during the rebuttal period, and has shown the advantage of our algorithm in practice. Moreover, it is also worth noting that our algorithm has a potential application in memory-efficient fine-tuning of large language models (LLM). Very recent studies [1,2] have utilized zero-order optimization algorithms to reduce the memory required by fine-tuning LLM. Our algorithm may be utilized to further achieve the asynchronous update, because of its ability to deal with delayed feedback. We will provide more discussions about our potential applications in the revised version.
>
> [1] Y. Zhang et al. Revisiting Zeroth-Order Optimization for Memory-Efficient LLM Fine-Tuning: A Benchmark. In ICML, 2024.
>
> [2] S. Malladi et al. Fine-Tuning Language Models with Just Forward Passes. In NeurIPS, pages 53038–53075, 2023.
>
> ---
> Q2: Could the authors comment on what the problem looks like for the regular stochastic case with all $f_t=f$ for some fixed $f$, and how this might make the problem easier under delays?
>
> A2: We notice that the stochastic case of delayed online convex optimization (OCO) has been investigated in the pioneering work of Agarwal and Duchi [3], which is referred to as distributed or parallel stochastic optimization with asynchronous update. Therefore, the stochastic case of delayed bandit convex optimization (BCO) reduces to a zero-order variant of this asynchronous optimization problem. Moreover, as discussed in Agarwal and Duchi [3], the stochastic case itself is not sufficient to make the delayed problem easier. Actually, the smoothness assumption on the loss functions is also required.
>
> More specifically, Agarwal and Duchi [3] show that in the stochastic case, the delay only increases the regret of delayed OCO for convex and smooth functions in an additive way, i.e., an $O(\sqrt{T}+d^2)$ regret bound. The key insight for this improvement is that the perturbed error of the delayed stochastic gradients can be much smaller under the smoothness assumption. Therefore, it is natural to conjecture that the stochastic case will make delayed BCO for convex and smooth functions easier in a similar way. We will provide detailed discussions about the stochastic case in the revised version.
>
> [3] A. Agarwal and J. Duchi. Distributed Delayed Stochastic Optimization. In NIPS, pages 873–881, 2011.
>
> ---
> Q3: Why do the authors conjecture that the lower bound depends on $\bar{d}$, whereas the upper bound depends on $d$? What improvements would be required to get these to match?
>
> A3: Our conjecture of the dependence on $\bar{d}$ stems from the existing $\Omega(\sqrt{\bar{d}T})$ lower bound [Bistritz et al., 2022] for delayed BCO. However, as discussed in our paper, it is hard for our algorithm to achieve regret bounds depending on $\bar{d}$ due to the blocking update mechanism. Nonetheless, we want to emphasize that our regret bounds achieved via the blocking update mechanism are already quite favorable, and it is highly non-trivial to develop a better algorithm, especially in the case with strongly convex functions. Note that even in the easier full-information setting, previous studies can only achieve the $O(d\log T)$ regret for strongly convex functions, which is the same as the delay-dependent part in our regret bounds for strongly convex functions.

---

> > ### Comment · Reviewer_vCin · 2024-08-09
> > **Thanks for the clear response**
> >
> > Thank you for the response, which I find quite helpful. I will stick with my positive score.

---

### Author Rebuttal · Authors · 2024-08-04

## Common Response to the Suggestion about Experiments

We thank all the reviewers for your detailed comments. In the following, we first respond to the common suggestion about experiments, and other questions are addressed in a separate response for every reviewer. Please let us know if you have any further questions.

During the rebuttal period, we conducted experiments on two publicly available data sets—ijcnn1 and SUSY from the LIBSVM repository [1]. Specifically, we randomly select $T=40000$ examples from the original data sets. The dimensionality of ijcnn1 and SUSY are $n=22$ and $n=18$, respectively. Moreover, we consider online binary classification over a convex set $\mathcal{K}=\\{\\|\mathbf{x}\\|_2\\leq 50\\}$. In each round $t\in[T]$, the adversary chooses the hinge loss $f_t(\mathbf{x})=\\max\\{1-y_t\\mathbf{w}_t^\\top\\mathbf{x},0\\}$, where $\mathbf{w}_t$ and $y_t\in\\{-1,1\\}$ are the feature vector and class label of the $t$-th example, respectively.

Different values of the maximum delay $d$ in the set $\\{200, 600,1000,\dots, 5000\\}$ have been tried in experiments on both data sets. For each specific $d$, to simulate arbitrary delays, $d_t$ is independently and uniformly sampled from $[1, d]$. Note that, in this way, the average delay satisfies $\mathbb{E}[\bar{d}]=\frac{d+1}{2}$, and thus is close to the maximum delay $d$. We compare our D-FTBL against GOLD [Héliou et al., 2020] and improved GOLD [Bistritz et al., 2022]. Due to the randomness of these algorithms, we repeat them 20 times and report the average of their total loss.

Figure 1 in the attached PDF shows the numerical results, and we have the following main observations.
1) For our D-FTBL, when $d$ increases from $200$ to $5000$, the growth of the total loss is very slow, which is consistent with the dependence of our regret bound on $d$. Note that $d=5000$ is larger than $n\sqrt{T}$ in our experiments.
2) From $d=600$ to $d=5000$, the total loss of our D-FTBL is better than both GOLD and improved GOLD, which verifies the advantage of our algorithm in the delayed setting.
3) Although for $d=200$, D-FTBL is slightly worse than baselines, it is reasonable because the block update mechanism enlarges each delay to be at least the block size, which could result in a slightly larger constant factor in the regret.

[1] C.-C. Chang and C.-J. Lin. LIBSVM: A library for support vector machines. ACM Transactions on Intelligent Systems and Technology, 2(27):1–27, 2011.

---

### Decision · Program_Chairs · 2024-09-25

**Decision:**

Accept (poster)

**Comment:**

This work studies the BCO problem with delayed feedback and presents a novel FTRL algorithm that uses a blocking update mechanism to decouple the effect of the delayed feedback on the regret of the algorithm. Most reviewers find the papers presentation to be very good, that the proofs are sound and that the authors have done a good job with literature review and comparison to prior work. Most reviewers also find the blocking update idea to be novel and interesting and point out that it could have application in other problems. One reviewer has suggested that the authors include an empirical evaluation of their algorithm which the authors do in the rebuttal. The only raised concern during the discussion phase is that of how impactful the algorithm or technical contribution of this paper are, given that the analysis builds on the standard FTRL framework. I find that the blocking update is interesting enough by itself, as other reviewers have also pointed out, and that the technical contributions are good enough for NeurIPS. As such I would recommend this paper for acceptance. I encourage the authors to incorporate the empirical evaluation in the final version of the paper.